

# Seasonal variability of the circulation in the Arabian Sea at intermediate depth and its link to the Oxygen Minimum Zone

Henrike Schmidt[1,2], Rena Czeschel[1], Martin Visbeck[1,2]

[1]GEOMAR Helmholtz Centre for Ocean Research Kiel, Düsternbrooker Weg 20, 24105 Kiel, Germany
[2] Kiel University, Christian-Albrechts-Platz 4, 24118 Kiel, Germany

*Correspondence to*: H. Schmidt (hschmidt@geomar.de)

**Abstract.** Oxygen minimum zones (OMZs) in the open ocean occur below the surface in regions of weak ventilation and high biological productivity. Very low levels of dissolved oxygen affect marine life and alter biogeochemical cycles. One of
the most intense but least understood OMZs in the world ocean is located in the Arabian Sea in a depth range between 300 to 1000 m. An improved understanding of the physical processes that have an impact on the OMZ in the Arabian Sea is necessary for a reliable assessment of its current state and future development.

This study uses a combination of observational data as well as reanalysis velocity fields from the ocean model HYCOM (Hybrid Coordinate Ocean Model) to investigate the advective pathways of Lagrangian particles into the Arabian Sea OMZ
at intermediate depths.

In the eastern basin, the OMZ is strongest during winter monsoon with a core thickness of 1000 m depth and oxygen values of less than 5 μmol/kg. The minimum of oxygen concentration might be favoured by a maximum advection of Lagrangian particles that follows the main advective pathway along the perimeter of the basin into the eastern basin of the Arabian Sea during winter monsoon. These Lagrangian particles pass regions of high primary production and respiration contributing to a
transport of low oxygenated water into the eastern part of the OMZ.

The maximum of oxygen concentration in the western basin of the Arabian Sea in May coincides with a maximum southward advection of particles along the western boundary during spring intermonsoon supplying the western core of the OMZ with higher oxygenated water.

The maximum of oxygen concentration in the eastern basin of the Arabian Sea in May might be associated with the
northward inflow of Lagrangian particles across 10° N into the Arabian Sea which is highest during spring intermonsoon.

The Red Sea outflow of advective particles into the western and eastern basin starts during the summer monsoon associated with the northeastward current during the summer monsoon. Whereas particles from the Persian Gulf advect over the whole year.





As the weak seasonal cycle of oxygen concentration in the eastern and western basin can be explained by seasonal changing advective pathways at intermediate depths into the ASOMZ, the simplified backward trajectory approach seems to be a good method for prediction of the seasonality of advective pathways of Lagrangian particles into the ASOMZ.

## 5  1 Introduction

Oxygen concentration at intermediate depth levels in the ocean is a result of the transport of oxygen from the surface mixed layer into the ocean interior (ventilation) and the local consumption of oxygen in the context of biological productivity and sinking organic matter (microbial respiration). In the south eastern parts of the tropical ocean poor ventilation south of the subtropical gyre circulation (Luyten et al., 1983) and in the northern Indian Ocean because the lack of ventilation from the

north combined with high biological production in upwelling regions results in enhanced oxygen consumption by sinking organic matter and consequently very low levels of dissolved oxygen below the surface (Stramma et al., 2008; Gilly et al., 2013). These regions, so called oxygen minimum zones (OMZ) are characterized by low oxygen concentrations spanning a depth range of about 200-700 m depth (e.g. Karstensen et al. 2008).

It is well established that OMZs affect marine biogeochemical processes such as the global carbon and nutrient cycles

(Bange et al., 2005; Naqvi et al., 2006). Conditions in near complete oxygen-depleted upwelling regions favour denitrification, which enhances production and release of climate-relevant trace gases to the atmosphere (Naqvi et al., 2010; Shenoy et al., 2012). Moreover, OMZs build a respiratory barrier in the subsurface layer impinging the ecosystem structure by limiting suitable habitats (Levin et al., 2009; Stramma et al., 2012; Resplandy et al., 2012).

Observations as well as global and regional models show a global trend towards decrease of oxygen and spatial expansion

and intensification of OMZs during the last decades with noticeable regional variations (Stramma et al., 2008, 2010; Keeling et al., 2010; Diaz and Rosenberg, 2008). Declining oxygen is anticipated to intensify especially in coastal regions in response to global warming (Keeling et al., 2010; Schmidtko et al., 2017), which affects changes in ventilation, stratification, and solubility as well as eutrophication causing microbial respiration (Diaz and Rosenberg, 2008; Keeling et al., 2010, Breitburg et al., 2018). Therefore, deoxygenation has become another major stressor affecting the marine and climate system besides

warming and acidification and evolves into an important indicator for a changing oceanic environment.

Although there is no precise threshold where macro-organisms experience stress or die, or chemical cycles switch to alternative pathways the community has established four oxygen regimes and approximate thresholds. The boundary between oxic and hypoxic conditions is defined at 60 µmol/kg (Gray et al., 2002; Keeling et al., 2010). Regimes are termed suboxic if oxygen concentration drops below 10 µmol/kg (Keeling et al., 2010) and nitrate involved respiration begins

(Bange et al., 2005). Regions are called anoxic when dissolved oxygen drops below a few µmol/kg and sulphate reduction is the dominant respiratory process (Naqvi et al., 2010).





In this study we focus on the Arabian Sea OMZ (ASOMZ). It has the smallest horizontal extent of all open ocean OMZs, but is one of the most intense in the world tropical ocean based on the largest vertical extent of hypoxic water (Kamykowski and Zentara, 1990) as well as on a significant core thickness with suboxic conditions of oxygen concentrations below 3 µmol/kg (Rao et al., 1994). Observations reveal an intensification of the northern part of the ASOMZ over the period of the last three

(Queste et al., 2018) to five decades (Ito et al., 2017) and a shoaling of the hypoxic boundary in the Sea of Oman (Piontkovski and Al-Oufi, 2015). The expansion of the ASOMZ is accompanied with declining sardine landings and an increase of fish kill incidents along the Omani coast (Piontkovski and Queste, 2016). Further expansion of the ASOMZ might have dramatic consequences on marine habitats and ecosystems (Keeling et al., 2010; Stramma et al., 2012). Hence food security and livelihoods of one of the most populous regions on earth - about 25% of the world's population lives in the

Indian Ocean rim countries - would be strongly affected (Breitburg et al., 2018). To understand ocean-climate interactions it is necessary to advance knowledge about the factors that impact the ventilation of the pronounced ASOMZ as e.g. water mass advection and large-scale circulation.

While the circulation of the upper-ocean is fairly well known from drifter data (Shenoi et al., 1999) and satellite altimetry (Beal et al., 2013) precise subsurface ventilation pathways of water masses entering the AS beneath the surface layer are less

well understood in detail due to a lack of observational data (McCreary et al., 2013) and the complex interactions with the monsoon cycles. One unique difference of the Indian Ocean OMZs compared to the other ocean basins, that host OMZs is the fact that the upper layer of the Indian Ocean in general and the Arabian Sea (AS) in particular are strongly impacted by the Asian monsoon system resulting in a seasonally reversal of all boundary currents and associated ocean ventilation patterns. Monsoonal wind forcing enabled by the land boundary in the north shifts from southwest winds during summer

monsoon, causing strong upwelling off the coasts of Somalia and Oman, to northeast winds during winter monsoon driving downwelling circulation (Schott et al., 2001). The seasonal changes significantly influence biogeochemical cycles, biological activity and ecosystem response (Hood et al., 2009; Resplandy et al., 2012; Brewin et al., 2012).

Ventilation in the OMZ layers of the AS is facilitated by three major intermediate source water masses. Oxygenated Indian Central Water (ICW) enters the AS at intermediate depth from the south (Fig. 1). High salinity Persian Gulf Water (PGW)

enters the AS just beneath the thermocline in the north, spreading southward as well as along the perimeter of the basin (Prasad et al., 2001). Low salinity but denser Red Sea Water (RSW) enters the AS at intermediate depth and spreads across the basin (Beal et al., 2000; Shankar et al., 2005).

Several assumptions were made to explain the dynamical and biological processes associated with the shape of the ASOMZ. So far it is known that, unlike in other tropical ocean basins, slow advection time is not responsible for the maintenance of

the OMZ in the AS, where low-oxygen water has a residence time of 10 years (Olson et al., 1993). According to Sarma (2002) the residence time is even shorter with 6.5 years and the maintenance of the OMZ is caused by sluggish circulation combined with biological varying activities. Other studies explained the relatively high oxygen rates at the western boundary with the supply of oxygen-rich water transported by the western boundary current (Swallow, 1984; Sarma, 2002) and by mixing of mesoscale eddies (Kim et al., 2001), although this is a region with high primary production at the surface and





associated high consumption rates below. A process study of McCreary et al. (2013) stated the importance of the large-scale circulation for the shape of the ASOMZ as well as mesoscale features for variations of dynamical and biological processes. Several studies have simulated the Indian Ocean circulation, whereby current model systems reveal large uncertainties and differences amongst them (McCreary et al., 2013). Typically, coarse resolution coupled biogeochemical ocean models

exhibit strong biases and tend to simulate lower oxygen concentrations in the Bay of Bengal than in the AS (e.g. Oschlies et al., 2008) contradicting the existing observations. Lachkar et al. (2016), however, suggests that the model performance improves with increasing model resolution. The latter findings support suggestions by Resplandy et al. (2012) and McCreary et al. (2013) that horizontal eddy mixing strongly impacts the oxygen dynamics in the AS. Studies on the equatorial Pacific from global coupled biogeochemical circulation models (Dietze and Löptien, 2013) point out that poor model performance is

related to a deficient representation of ventilation pathways rather than being associated with a deficient representation of biogeochemical processes (i.e. respiration). This confirms the need for a better understanding of the intermediate circulation and it's seasonality in the AS including the pathways of RSW and PGW to understand the associated variability of the ASOMZ and related climate-biogeochemical interactions.

It remains an open question how the interplay between physical and biogeochemical processes influences the Indian Ocean

oxygen dynamics. Specifically, there are two issues for the ASOMZ:

1) Why does the ASOMZ occur further east relative to the upwelling area with associated high productivity? A good explanation of this eastward shift could not be given so far (Acharya and Panigrahi, 2016).

2) Why is the ASOMZ maintained throughout the year with only a weak seasonal cycle compared to the dramatic changes of physical forcing and biogeochemical conditions associated with the seasonal reversing monsoon winds?

Therefore, the present study focuses on advective pathways relevant for the ventilation dynamics of the ASOMZ. Main experiments are focused on the circulation on the isopycnal layer of $\sigma = 27$ kg/m$^3$ which is associated with the upper core of the ASOMZ. A backward-trajectory analysis was applied to examine the source regions of the seasonally changing advective pathways of the major water masses in the ASOMZ. For a better understanding of the zonal gradient of the ASOMZ we calculated the pathways of Lagrangian particles based on two release points that are located in the eastern and

the western basin of the Arabian Sea.

The following section explains data sets as well as design of the experiments and methods used for this study. Section 3 presents main ventilation pathways for the eastern and western basin of the AS, as well as their time scales and seasonality that are relevant for the variability of the ASOMZ and associated uncertainties. This is followed by the discussion and conclusions in section 4.

**2 Data and methods**

2.1 Data sets





The study uses the global dissolved oxygen climatology of the World Ocean Atlas 2013 (WOA13) as observational data. The monthly mean data cover a period from 1955-2012 and are available with a spatial resolution of 1° x 1° interpolated on 102 depth levels (Garcia et al., 2013). Trajectory calculations are based on reanalysis velocity data from the dynamic ocean model HYCOM (Hybrid Coordinate Ocean Model) (Bleck, 2002) provided by the Center for Ocean-Atmospheric Prediction

Studies (COAPS). The model has a spatial resolution of 1/12° in longitude and latitude with 40 depth levels between 0 and 5000 m, with decreasing resolution towards greater depth from 2-1000 m. It has a realistic bathymetry based on the General Bathymetric Chart of the Oceans (GEBCO) and uses isopycnal coordinates in the open, stratified ocean, changes to terrain-following coordinates in shallower and coastal regions and uses z-level coordinates in the mixed layer. The surface forcing from the National Center for Environmental Prediction (NCEP) Climate Forecast System Reanalysis (CFSR) is used in the

time between 1995-2012. Furthermore, HYCOM is run in data assimilation mode using gridded 'observations' from the Navy Coupled Ocean Data Assimilation (NCODA) system (Cummings, 2005; Cummings and Smedstad, 2013).
The variable vertical coordinates are beneficial for the model to better reproduce the circulation near out-/overflow regions compared to typical z-level models, which would generally have problems to resolve the shallower coastal regions properly (see also Bleck and Boudra, 1981; Bleck and Benjamin, 1993; Bleck, 2002). The latter is important for the analysis of the

supply of Persian Gulf Water (PGW) and Red Sea Water (RSW) through the Gulf of Oman and the Gulf of Aden respectively.
The model velocity output available to use were daily snapshots for the time period from January 2000 to December 2012. The velocity field during winter and summer monsoon is shown using the mean seasonal velocity for the months November to February and June to September, respectively, averaged for the years 2000 to 2012. The data are spatially filtered using a

0.6° x 0.6° window and presented on a grid with the same resolution (Fig. 2).
For a validation of the HYCOM velocity data we compared the near-surface circulation of HYCOM with the climatology of YoMaHa'07, which is based on observational data obtained from Array of Real-time Geostrophic Oceanography (ARGO) floats (Lebedev et al., 2007). This choice is motivated by the lack of observational data at intermediate depths in the Arabian Sea. The comparison of the near-surface circulation during the winter and summer monsoon between HYCOM and ARGO

agrees very well which is given in Fig. S1 in the Supplement. The complex circulation pattern at the near-surface which is strongly affected by the seasonal Asian monsoon is well described by the HYCOM data reflecting all (reversing) currents that are relevant for the AS.

### 2.2 Design of the experiment

In the following we would like to present motivation and key details of the conceptual design of the experiments to investigate the main advective pathways within the OMZ of the AS and its seasonal variability. To estimate the advective contribution to the OMZ ventilation we analysed the pathways of Lagrangian particles released into a two dimensional (along isopycnal) model based velocity field. This approach is time efficient, focusses on the isopycnal advection but ignores





for example the effects of upwelling or diapycnal mixing. However, this method allows both to estimate contribution from different source regions to the OMZ by performing backward trajectories and to draw inferences of the basin wide spread of oxygen at intermediate levels. Our experiments are based on the assumption that PGW and RSW are the main local source water masses that are relevant for the ventilation of the ASOMZ and that the oxygen rich waters follow largely their

isopycnal layer horizontally. Therefore, advective pathways from Lagrangian particles into the OMZ are calculated on an isopycnal associated with the source regions of the PGW and RSW as well as the OMZ core region. A good representative isopycnal surface of 27 kg/m³ was chosen for most experiments based on two main reasons: The isopycnal lies in the upper core of the ASOMZ (Fig. 3) with low oxygen values of less than 10 $\mu$mol/kg nearly throughout the entire year (Fig. 4a). Furthermore, this is the density layer with seasonal changes in oxygen concentration (Fig. 4a). The core densities of PGW ($\sigma$

= 26.4 kg/m³) and RSW ($\sigma$ = 27.4 kg/m³), which appear to be the main source water masses ventilating the ASOMZ, bracket the isopycnal density of $\sigma$ = 27 kg/m³. The isopycnal density layer on which ICW is advected northward ($\sigma$ = 26.7 kg/m³) lies between the ones of RSW and PGW. For the AS, the supply of oxygen was suggested by Banse et al. (2014) to be on the isopycnal surfaces of 27 kg/m³ at depths between 300-500 m depth.

The contrast in extension and seasonal cycle, not only in oxygen but also in biogeochemical activity (Hood et al., 2009; Resplandy et al., 2012; Brewin et al., 2012), of the ASOMZ for the eastern and western basin encourages to analyse the ventilation of each half of the basin individually. Therefore, we define two release locations in the eastern (ER) and western (WR) part of the core of the ASOMZ (Fig. 1). The western part is associated with the area of high primary production and the eastern part is associated with the area of lowest oxygen values. Both release locations represent the core of the OMZ

and are defined as circles with a radius of twice the grid spacing, thus 1/6°, around the launching coordinates, which are 19.04° N and 66.64° E for the ER and 19.04° N and 62.00° E for the WR. The Lagrangian particles are spread equally over that area and are all released at the same time (for one run). For the forward trajectories two additional release locations in the Gulf of Aden, simulating the spreading of Red Sea Water (RS, 49.04° E and 13.04° W) and in the Gulf of Oman, simulating the spreading of Persian Gulf Water (PG, 59.04° E and 24.00° N; Fig. 1) were chosen.

After analysing the pathways of Lagrangian particles within the ASOMZ (Section 3.2) we focused on the seasonal variation of the circulation in the AS (Section 3.3). To address this question we chose distinct sections along the main advective pathways of the Lagrangian particles (Fig. 1) and calculated the transit times of the particles to get from one region to another for different pathways: two zonal sections are at equal distance south and north of the release

locations (17° N, 21° N) to investigate the impact of the northeast and southwest monsoon on the advection of the particles, a meridional section separates the eastern and western half of the basin between the release locations at 64.3° E to determine the interior circulation, two meridional sections are located at the borders to the Gulf of Oman and Gulf



of Aden as the source of the main water masses and another zonal section at 10° N serves as a southern boundary of the AS as our research area to get an insight of the inflow from the south and its variation.

The Lagrangian particles were advected using the two dimensional velocity fields from HYCOM reanalysis velocity fields following basic relations of continuous deformation (see Supplement, Lamb, 1879). This approach is consistent with more recent techniques as described in van Sebille et al. (2018).

The daily velocity fields were vertically linearly interpolated onto the target isopycnal surface. The number of Lagrangian particles released is 50000 for the runs that were mainly used for statistical purpose (see also Section 2.4) and 10000 for runs 1 to 10 (Tab. 1). The particles were advanced using an Euler forward-in-time integration scheme using a time step of 1/20 day. Both forward and backward trajectories were calculated and particle positions are stored every $4^{th}$ day. In addition to the model velocity field a random walk of particles is applied to represent subscale diffusion of 20 $m^2$/s. Near the coast a special case of random walk in the offshore direction is used to prevent trajectories leaving the ocean. The choice of magnitude of random walk is connected to the spatial and temporal grid resolution. A sensitivity experiment with different subscale diffusion coefficients of 10, 20 and 25 $m^2$/s does not reveal significant different results (not shown here). Nevertheless, there are some grid boxes along the coastline and especially near islands, where the particles get trapped. These spuriously high probabilities were not considered for further analyses. Moreover, the velocity fields of HYCOM are obviously divergent, in particular in up and downwelling regions near coasts and islands (e.g. the Maledives, Socotra).

Several sensitivity runs were conducted many of them with a reduced number of particles to save computational costs. A comparison between full and reduced number of particles gave very similar results (not shown here). In order to estimate the representativeness of the main Lagrangian pathway analysis on the 27 $kg/m^3$ isopycnal surface two further runs on a shallower and deeper isopycnal were done (for PGW σ = 26.4 $kg/m^3$ and RSW σ = 27.4 $kg/m^3$). These experiments also used repeated daily velocity data for the calendar year 2006 for costs savings.

To estimate the impact of slower diffusion effects on the ventilation of the ASOMZ we compared the typical 8 year long results (Tab.1) to longer 13 year model runs. Again, both experiment gave very similar results pointing towards a secondary role of the slower processes.

### 2.3 Trajectory visualization

To analyse the Lagrangian data, the AS is divided into a grid of 1° x 1° resolution. For each time step the number of particles residing in a certain grid box can be counted leading to a map that shows the particle concentration over the analysed time in a certain grid box or at individual time steps (Gary et al., 2011). For a better comparison, the probability for each bin has been obtained. Summing up all particle counts in a certain grid box over the whole time and dividing it by the total number



of particle counts for all grid boxes leads to probability maps that sum up to 100% for the whole experimental area and time (van Sebille et al., 2018).

With a subsample of the trajectories that reach the source regions, these maps can highlight the most likely advective pathways of Larangian particles (Gary et al., 2011). Additionally, it is possible to analyse the spreading of the particles by

looking at single time steps.

The point to point transit time describes the time that each individual Lagrangian particle takes to transit between defined regions (van Sebille et al., 2018). The transit time is analysed along identified main advective pathways into the ASOMZ between distinct sections (see Section 2.2). The transit time is not unique, as different Lagrangian particles might travel between two regions on different ways in different length of time (Phelps et al., 2013). The here discussed transit times are

thus defined by the times where 50% of the particles crossed the distinct sections (percentages refer to the total number of Lagrangian particles that have crossed the section after the whole time span of the simulation (8 years). Therefore, no particle is counted twice as only the first crossing time of each particle at each section is detected. Additionally, the seasonal cycle of Lagrangian particles crossing these sections can be determined.

**2.4 Trajectory validation and statistics**

To test the reliability of the calculated Lagrangian trajectories 5 model runs with identical setup were performed (each with 50000 particles, 13 years duration, starting all at the ER in December 2012). The differences between these runs are discussed in Section 3.5.

To detect the interannual variability the runs with the duration of 13 years used for the statistics were compared to a

climatological run, which was performed with velocity fields with the mean daily velocity of the 13 years at each grid point and day. Furthermore, the 8 year long runs (runs 1- 6; Tab. 1) were started with a temporal offset of 2 years between the individual runs. For the analysis of seasonality and transit time, we used the mean of these runs to smooth out the interannual variability.

Seasonal differences in particle movement around the release locations can be predicted by starting the calculations with a

lack of 3 month (January, April, July and October). This was done for forward calculated trajectories from the RS/PG release to predict the spreading op RSW/PGW.





## 3 Results

### 3.1 Seasonal oxygen dynamics and circulation at intermediate depth

The northern Indian Ocean is a region of strong monsoonal forcing and it is known that seasonal changes have a profound impact on the ASOMZ (Resplandy et al., 2012). Thus, first we give a short overview of the seasonal variability of the

suboxic oxygen distribution and the circulation at intermediate depth in the AS.

The OMZ in the northern AS has regional differences in the seasonal cycle, especially in the upper core (350 – 550 m depth, Fig. 4). The annual mean of oxygen from observational climatologies shows that the layer containing oxygen of less than 10 µmol/kg is deepest in the eastern basin (Fig. 5a). The maximum thickness arises during fall intermonsoon and at the beginning of winter monsoon with a depth of 1000 m (Fig. 4b) and nearly total oxygen depletion in the core (Fig. 4a).

Oxygen concentration increases within spring intermonsoon and the suboxic layer in the eastern AS nearly vanishes in May (Fig. 4b).

A similar seasonal cycle is prominent in the western AS with a maximum thickness of the suboxic layer of 900 m (Fig. 4b) but however, a weaker ventilation during the spring intermonsoon compared to the east. The layer containing oxygen of less than 10 µmol/kg remains thicker than 500 m. Based on an area of 2°x 2° in total centered around the release location the

spatial standard deviations were calculated (Fig. 4). They show that the seasonal cycle of the OMZ represents a large area and not only the release location. This holds especially for the eastern basin.

Lagrangian trajectories were calculated on basis of the daily HYCOM reanalysis velocities on the isopycnal surface of 27 kg/m³. Also at intermediate depth in the AS several boundary currents are seasonally changing directions such as the Somali Current along the western boundary. Mean seasonal velocity for the period of 13 years shows the reversing of the Somali

Current from a southwestward boundary current during winter monsoon (Fig. 2a) to a stronger northeastward boundary current during summer monsoon (Fig. 2b). The annual mean of the Somali Current reveals a northeastward boundary current. Generally, velocities show the strongest variability along the boundaries, especially in the western basin increasing towards the equator (Fig. 2c) and in the marginal seas. Along the eastern boundary of the AS the flow at intermediate depth is also changing its direction between the different monsoon phases from a distinct southeastward directed flow along the

west coast of India during the northeast monsoon (Fig. 2a) to a northwestward directed more variable flow during the southwest monsoon (Fig. 2b).

### 3.2 Particle origins and main pathways

The presented main advective pathways of Lagrangian particles at intermediate depth within the AS are based on backward

trajectories calculated for a time span of 8 years. Release points are located in the eastern and western part of the ASOMZ and calculations were done on three different isopycnal levels.

On their way into the eastern part of the ASOMZ most of the virtual Lagrangian particles follow the north- and southward advective pathways along the eastern boundary of the AS. For the ER runs we find the highest particle probability along the North Indian and Pakistani coastline on all three isopycnal levels (Fig. 6b, S3b, d). The northward advection of particles along the coast of India from the southeastern part of the AS is especially pronounced on the isopycnal level of $\sigma = 27$ kg/m$^3$

(Fig. 6b). A similar pattern but with lower probability is also seen on the isopycnal level of $\sigma = 26.4$ kg/m$^3$ (Fig. S3b). Both pathways along the eastern boundary of the AS at intermediate depth are confirmed by the seasonal mean of the circulation for the winter and summer monsoon (Fig. 2a, b).

In the western part of the ASOMZ the highest probability of advecting particles occurs north of the release location on all three depth levels (Fig. 6a, S3a, c) but more equally spread around the release location compared to the eastern basin. This is

also reflected in the high variability of the velocity field in the northwest corner of the AS (Fig. 2c). A pattern with lower probability extends in a southwest to northeast direction along the western boundary pointing towards southward particle advection from the Gulf of Oman and the PG during the winter monsoon and northward particle advection from the RS into the western ASOMZ (Fig. 6a) during the summer monsoon.

The pattern of particle distribution after a simulation of 4 years (see supplement Fig. S2b) confirms the particle distribution

along the west coast of Indian and Pakistan seen in the patterns over the whole time series (Fig. 6b). Simulations of another 4 years backward show a wider and more equally spread distribution of particle origins that ventilate the eastern part of the ASOMZ (Fig. S2d).

The snapshot of the particle distribution after 4 years shows the source of advecting particles also from the eastern part of the ASOMZ (Fig. S2a) for the WR. Similar to the ER, the particle origins after 8 years of simulation spread wider and more

equally over the AS (Fig. S2c). After 12 years of simulation the origins of advected particles reveal no fundamental differences between the WR and ER (Fig. S2e, f). Therefore, in the following we consider the main advective pathways based on backward trajectories calculated for a time span of 8 years.

For two of the three major intermediate source water masses that ventilate the ASOMZ - the RSW and the PGW - the source area can be localized clearly so that it is possible to extract the particles that origin from the RS and the PG flowing through

the Gulf of Aden and the Gulf of Oman, respectively. The most prominent advective pathway of Lagrangian particles from the PG and the RS circles the basin clockwise along the western, northern and northeastern boundary into the eastern ASOMZ (Fig. 7b, d). RSW spreads mostly northeastward along the coast of Yemen/Oman, where it enters the western part of the ASOMZ (Fig. 7c). Most particles spread further north along the coastline of Pakistan and India to enter the eastern basin, whereas the more direct interior pathway is less frequent (Fig. 7d). The pathway of PGW into the western part of the

ASOMZ is directed southward along the coastline off Oman (Fig. 7a).

Water entering from the south at intermediate depth (ICW) shows a direct interior exchange from the eastern to the western basin in the region of the OMZ (Fig. S6). It is also noteworthy that the particles enter the AS more frequent from the southeast and tend not to follow the western boundary current on the isopycnal level $\sigma = 27$ kg/m$^3$. The pathways of virtual Lagrangian trajectories are confirmed by the velocity fields at intermediate depths revealing prevailing northward currents in





the eastern basin at about 10°N during the southwest monsoon (Fig. 2b) and northeastward currents in the western and central basin at about 10°N during the northeast monsoon (Fig. 2a).

The pattern of advective pathways of Lagrangian particles associated with RSW and PGW calculated on the isopycnal level $\sigma = 27$ kg/m$^3$ (Fig. 7) can be overall confirmed by the trajectories calculated on the shallower ($\sigma = 26.4$ kg/m$^3$) and deeper ($\sigma$

$= 27.4$ kg/m$^3$) isopycnal levels (Fig. S4, S5). Broader pathways on isopycnal $\sigma = 26.4$ kg/m$^3$ with lower values of particle probability point towards a stronger mixing in more shallow depths due to the monsoon (Fig. S4). It is also noteworthy that the pathway of RSW to the eastern basin on the isopycnal surface of $\sigma = 27.4$ kg/m$^3$ is more frequent in the interior of the AS than the advective pathway along the perimeter in the northern AS (Fig. S5d) in opposite to the circulation on isopycnal $\sigma = 27$ kg/m$^3$ (Fig. 7d).

### 3.3 Particle transit time and percentage

After analysing the main pathways, in the following we focus on the point to point transit time of the advective Lagrangian particles helping to further understand the circulation at intermediate depth. Therefore, the point to point transit time of the

particles across selected sections along their distinct pathways (see section 2.2 for location of the sections) is analysed. As transit time is individual for each particle Fig. 8 shows the cumulative transit time of all particles crossing that section on the isopycnal surface of 27 kg/m$^3$. Additionally, the times where 50% of the particles crossed the distinct sections (percentages refer to the total number of Lagrangian particles that have crossed the section after the whole time span of the simulation (8 years)) are listed in Tab. 1.

The western part of the ASOMZ is ventilated preferably from particles coming from the northern basin. Within the first year about 60% of the released particles travel the pathway northward along the western boundary between the 21° N section and the WR (Fig. 8a). The number of particles travelling northward over the section at 17°N is much smaller. Barely 5% of all particles cross that section during the whole calculation time (Fig. 8a).

In contrast, in the eastern basin the numbers of particles ventilating the eastern part of the OMZ over the northern and the

southern section are about the same with rates of 52% (17°N) and 62% (21°N) of the released particles crossing over the 8 years of calculation time (Fig. 8b). Compared to the western basin (Fig. 8a) the slope of the cumulative particle transit time curve is flatter (Fig. 8b). Thus, the point to point transit times of the individual particles are spread over a wider and longer time range for the ER.

28% of the released particles are travelling around the perimeter of the basin (Fig. 8b), which is roughly 10% more than

particles taking the interior pathway between ER and WR (Fig. 8d). However, the exchange between WR and ER in the other direction is more pronounced (Fig. 8c). The point to point transit times for all these sections are less than six month for the fastest particles and the slope of the cumulative transit time is weak, especially for particles released in the eastern ASOMZ (Fig. 8b, d), pointing towards large differences in transit times for individual particles.



Transit times from the PG as well as from the RS are shorter to the western basin (Fig. 8c) than to the eastern basin (Fig. 8d, Tab.1). The mean transit time of 50% of the particles on the isopycnal surface of 27 kg/m$^3$ between the WR and the section in the Gulf of Oman is 2 years (for values of the single runs see Tab. 1). The equivalent mean transit time for the ER is 4.2 years. The mean point to point transit times from the release locations to the RS section for 50 % of the particles are 6.4 and

5.2 years for the ER and WR, respectively. Anyway, the slope of the curves is somehow constant over the whole calculation period, especially for the transit times to the RS (Fig. 8c, d).

The AS is also ventilated from the south across 10° N (Fig. 8e, f). For both release points, WR and ER, the ventilation is stronger across the eastern half of the basin (Fig. 8e, f). Here again, the slope of the curves is somehow constant over the whole calculation period. Mean transit times between the release locations and the south eastern section of the basin for 50

% of the particles are 4.8 and 6.0 years for the ER and WR and 5.4 and 5.6 years between the south western section and the ER and WR, respectively.

The point to point transit times at the isopycnal surface of 26.4 kg/m$^3$ are nearly entirely quicker compared to the ones that were discussed above for the 27 kg/m$^3$ isopycnal surface (Tab. 1). This tendency extends further down, as the deepest considered isopycnal surface of 27.4 kg/m$^3$ has the slowest transit times.

Due to a broad distribution, the number of particles that cross the sections decreases with the distance of the particles from their release point. This also means, that the remaining percentage of particles stays around the release location or in the adjacent area of one section.

### 3.4 Seasonal Variability

The seasonal variability of certain pathways is shown by the monthly average of the percentage of particles (Fig. 9). One of the strongest seasonal variability is revealed by the southward travelling Lagrangian particles between the sections at 21°N and the WR as well as the ER, however, with a more distinct amplitude in the western basin (Fig. 9a, b). The western basin is mainly advected by southward travelling particles from the northern part of the AS showing a distinct maximum during the intermonsoon phase in spring (Fig. 9a). This maximum coincides with the maximum in oxygen concentration in the

western basin (Fig. 4a) pointing towards a southward transport of higher oxygenated water during spring intermonsoon.

In contrast, in the eastern basin Lagrangian particles move preferably southward between the section at 21°N and the ER during the winter monsoon (Fig. 9b), which is also reflected in the southward eastern boundary current at intermediate depth (Fig. 2a). One of the most prominent pathways of the particles into the eastern basin, the eastward movement along the northern boundary across 64°E, is also strongest during the winter monsoon (Fig. 9b). Therefore, the minimum of oxygen

concentration in the eastern basin in winter (Fig. 4a) might be explained by the transport of lower oxygenated water due to the longer advection of particles while looping around the northern part of the basin and crossing regions with high primary production and resulting high consumption rates.



The advection of particles from the RS into the western basin increases with the beginning of the summer monsoon in July and peaks at the end of the summer monsoon in September (Fig. 9c). The northward transport along the western boundary is confirmed by the circulation at intermediate depth during the summer monsoon (Fig. 2b) and the seasonal cycle of the forward trajectories released in the RS (not shown). The same seasonal cycle is revealed for the advection from the RS into

the eastern basin, although weaker (Fig. 9d).

The transport of Lagrangian particles from the PG into the eastern basin shows a weak seasonal cycle, which peaks at the end of the winter monsoon (Fig. 9d). However, the transport of Lagrangian particles from the PG into the western basin (Fig. 9c) does not show a distinct seasonal cycle and so does the interior transport from the eastern OMZ into the western OMZ (Fig. 9c) reflecting the high variability in the AS. Anyway, the spreading of forward trajectories out of the PG reveals a weak

seasonal cycle of particles moving further into the mid AS and along the northern coast during summer monsoon, whereas during the winter monsoon particles stay closer to the western coast and travel southward (not shown).

  The eastward transport of particles along the northern boundary is weakest in spring inter-monsoon season (Fig. 9b). At the same time a higher direct interior transport from the western to the eastern half of the ASOMZ can be observed (Fig. 9d). The northward transport into the AS across 10° N mainly takes place at the eastern part of the basin showing a maximum of

the seasonal cycle during the spring intermonsoon (Fig. 9e, f). Hence, the maximum oxygen concentration at intermediate depth in spring (Fig. 4a) might be associated with the northward and eastward transport of higher oxygenated water into the eastern basin (Fig. 9b, d, f). The northward transport into the AS is weak at the western side of the basin in comparison to the eastern side and reveals a small cycle depending on the reversing monsoon (Fig. 9e, f).

**3.5 Trajectory error estimation and interannual variability**

One source of error arises from the calculation technique itself by adding a subscale diffusion and the random walk at the coastlines. To predict the discrepancy of that error, 5 runs were performed with identical setup (see Section 2.4). The percentage of trajectories reaching the PG/RS and southern IO and their mean transit times have standard deviations of 0.13, 0.01, 0.20/0.21 for PG particle percentage, RS particle percentage and ICW (east/west) particle percentage, respectively.

These differences are not distinguishable in the histogram maps of particle percentages.

As earlier discussed in Section 2.2 a different value for the diffusivity coefficient does not change the results significantly. Also, the reduction of number of released particles from 50000 to 10000 for the runs of 8 years (Tab.1) has no remarkable effect (Section 2.2).

Furthermore, Lagrangian particle probability maps for the simulations performed with climatological velocity values over a

duration of 13 years do not differ from the maps of runs that were performed with the continuous velocity data as described above. Nonetheless the particle percentages and transit times for the source regions differ. The highest discrepancies are a deviation of 30 % for the particles percentage of particles that travel from the PG to the western part of the ASOMZ. Other





runs show discrepancies between 4-6 % in particle percentages. A year-to-year time series analysis of the velocities shows strong damping for the climatology having peak velocities of 0.2 compared to about 0.8 m/s (not shown here).

To test the spatiotempotal variability, runs with the length of 8 years were performed with a temporal offset of 2 years between the runs. Again, the particle probability maps (Fig. 6) show similar results among each other (not shown here).

Concerning the standard deviations of particle percentages reaching the source regions, values lie between 0.3 for particles travelling between the RS and ER and 8.6 for particles travelling form the PG to the WR. These huge differences of particle amount from the marginal seas, as also the comparison with climatological velocity runs, let suspect a dependency on interannual variabilities, probably in the monsoon strength, influencing the intermediate circulation in the AS. Therefore, the seasonality analysis was performed with the 6 runs over 8 years to get a more confident result and better travel times,

smoothing out possible burst of years with strong ventilating currents.

Due to the small number of runs, it is not possible to give an estimation error, but the range in which order the values spread is given as discussed above.

**4 Discussion and Conclusions**

Focusing on the northwestern Indian Ocean, the strong seasonal cycle of the Asian Monsoon has an important impact on the

mixed layer of the Arabian Sea as well as on the circulation at intermediate layer. Between 450-500 m depth the western boundary circulation reverses direction from a strong northward flow in summer (Fig. 2a) to a weaker southward flow in winter (Fig. 2b). For the annual mean this sums up to a northward directed western boundary current, as previously found by Schott and McCreary (2001), with the strongest variability off the Somali and Omani coast (Fig. 2c).

During the summer monsoon enhanced upwelling occurs along the western boundary leading to a region of highest

biological productivity. Hence, the core of the ASOMZ is expected to be located at the same place at intermediate depth due to resulting high consumption rates below high productivity rates. However, the core of the ASOMZ is shifted away from that region and is more pronounced in the eastern basin than at the expected area along the western boundary (Fig. 5, 4; Acharya and Panigrahi, 2016). Suboxic conditions with oxygen concentration of less than 10 μmol/kg are found between 200 and 1000 m depth (Fig. 3) with weak annual variability in the deeper core according to gridded observational data (not

shown). The east-west contrast in oxygen concentrations found in the updated WOA 13 data (Fig. 4) confirms the results shown in Resplandy et al. (2012), who used a prior version of the WOA data.

However, oxygen concentrations from monthly mean gridded observations indicate a waek seasonal variability in the upper level of the ASOMZ showing higher oxygen values in spring intermonsoon (Fig. 4a), which are more pronounced in the east and slightly higher oxygen at the end of the winter monsoon only for the western basin. This ventilation goes along with a

shallowing of the suboxic layer in May and June (Fig. 4b). This seasonal variability was observed earlier by Sarma (2002) and Banse et al. (2014) showing higher oxygen values in the northern AS at around 300 m depth during the northeast monsoon.



In this study main pathways of virtual advective Lagrangian particles in the AS are determined on three different layers at the top, the middle and the bottom of the ASOMZ as well as their temporal and spatial variability. To consider the east - west contrast of the ASOMZ one release location of particles is placed in the eastern basin, where the suboxic layer is thickest during winter monsoon. The other release location is in the western part of the ASOMZ, where primary production
is strongest during the summer monsoon.

Present results from the trajectory calculations on the isopycnal density layer of 27 kg/m$^3$ reveal a main advective pathway of the eastern part of the ASOMZ along the perimeter of the basin (Fig. 7b, d). With the beginning of the summer monsoon, RSW spreads out of the Gulf of Aden (Fig. 9c) and flows northward along the coast of Oman, having the same direction as
the surface current (Fig. S1c). In the northwestern part of the basin where the Gulf of Oman merges with the AS, PGW about constantly runs out throughout the year (Fig. 9c, d). The eastward flow along the northern boundary and the southward flow into the eastern basin peaks during the winter monsoon (Fig. 9b). The long-distance advection of particles while looping around the northern part of the basin crossing regions with high primary production and resulting high consumption rates in the outflow region of the Gulf of Oman in winter (Acharya and Panigrahi, 2016; Lachkar et al., 2018) might lead to a
transport of lower oxygenated water which might be responsible for the minimum of oxygen concentration in the eastern basin in winter (Fig. 4a).

A more direct interior pathway, especially from the RS into the eastern basin south of 21°N is negligible confirming previous studies (Lachkar et al., 2016). Tracking the particles of RSW using a water mass analysis Acharya and Panigrahi (2016) reveals a maximal percentage of spreading along the coastlines but no propagation of RSW in the interior basin. However,
the direct interior exchange of water between the eastern and western interior part of the ASOMZ shows high variation, which is maximal in May (Fig. 9d). The interior pathway between the eastern and western basin for the exchange of watermasses becomes more pronounced on the deeper isopycnal surface of 27.4 kg/m$^3$ (Fig. S5d).

Particle probability maps (Fig. 6) also reveal the advection of particles from the southeast into the eastern part of the ASOMZ. The contribution of particles advecting into the eastern basin along the Indian coast is about similar from the north
and the south (Fig. 8b). Although, the surface currents (Fig. S1a, c) along the coast of south India reverse with the changing monsoon winds (Schott and McCreary, 2001), the northward transport at intermediate depth shows no clear seasonal cycle (Fig. 9b).

A more pronounced ventilation from the south (8°N) in the eastern AS was earlier found by Acharya and Panigrahi (2016). Even though ICW spreads northward uniformly across the basin at intermediate depth (You and Tomczak, 1993), our results
suggest that ventilating particles enter the AS predominantly along the eastern boundary (Fig. 8e, f; Schott and McCreary, 2001) in comparison to the western boundary, as they do in the thermocline (You and Tomczak, 1993). The maximum oxygen concentration at intermediate depth in May (Fig. 4a) can therefore be associated with the maximum northward transport of higher oxygenated water into the eastern basin during spring intermonsoon (Fig. 9f) and the supply of higher oxygenated water.



The western part advects particles equally from around covering particle origins in the northern AS. The RSW that spreads out of the Gulf of Aden during summer monsoon (Fig. 9c) passes the western basin OMZ on its way northward (Fig. 7c). These results are in agreement with the study of Beal et al. (2000), who tracked RSW spreading by salinity properties.

Compared to the strong variability of the western boundary current, the seasonal oxygen cycle in the western basin is weak (Fig. 4a). This could be explained by the loss of oxygen via consumption when RSW passes the area of strong primary production off the coast of Oman (Acharya and Panigrahi, 2016) during the summer monsoon.

Particle advection into the western basin from the north is strongest during the spring intermonsoon (Fig. 9a). This is confirmed by the maximum in oxygen concentration during May (Fig. 4a). However, Prasad et al. (2001) stated that PGW

spreads further down the Omani coast during winter monsoon with the western boundary undercurrent and more equally to the eastern basin around the northern pathway during the rest of the year. The same spreading patterns of PGW can be confirmed also on the isopycnal surface of 27 kg/m$^3$. The seasonal cycle of oxygen concentration shows a second peak in February (Fig. 4a).

Advective pathways from the marginal seas, which are bound to the western basin, are shorter to the western part of the

OMZ than to the eastern part, especially for water stemming out of the Gulf of Aden. The analysis of a point to point transit time of particles that reach the marginal seas shows that the mean transit time for 50% of the particles that travel between the PG and the ER is 4.2 years but just 2 years for the WR. Particles from the RS have a mean point to point travel time of 6.4 and 5.2 years to the ER and WR, respectively. However, prolonged transit times alone are not sufficient to explain the different characteristics in the eastern and the western part of the ASOMZ especially when considering the strong seasonal

variability of the advective pathways.

The comparison of travel times and particle percentages between different years (Tab. 1) as well as with climatological runs shows high discrepancies and standard deviations pointing towards a strong dependency on interannual variability, that is likely driven by the strength of the monsoon forcing.

Another point that underlines the connection between the monsoon forcing and both strength and variability of watermass

advection into the ASOMZ is the comparison between the results from 3 isopycnal layers (Tab. 1). With increasing depth the transit times become longer, pointing towards weaker currents and circulation.

However, to give true ventilation times or exact percentages of the water mass contribution to the ventilation more runs are required to calculate reliable statistics as well as extended time series are needed to confidently predict interannual variabilities. Nevertheless, the simplified backward trajectory approach seems to be a good method for prediction of the

seasonality of advective pathways of Lagrangian particles into the ASOMZ.

The seasonal changing advective pathways into the ASOMZ agree well with the weak seasonal oxygen cycle and show clear differences between the eastern and western basin. Thus we conclude that the water mass advection plays a crucial role for the eastward shift of the ASOMZ and might also be responsible for the maintenance of low oxygen throughout the year.





However, we cannot give a statement, how important it is compared to biogeochemical activities in the AS and their seasonality and impact on low oxygen values.

**Author contribution**

H. Schmidt, R. Czeschel, and M. Visbeck conceived the study. H. Schmidt handled all the data and performed the simulations. All authors discussed, wrote and modified the manuscript.

**Competing interests**

The authors declare that they have no conflict of interest.

**Data availability**

The 1/12° global HYCOM+NCODA Ocean Reanalysis output is publicly available at http://HYCOM.org. The WOA13 data are available at https://www.nodc.noaa.gov/OC5/woa13/woa13data.html.

**Acknowledgements**

Special thanks go to Professor Andreas Oschlies, my supervisor, who gave me time and support to continue this work during
my PhD. Financial support was received through GEOMAR. This work is a contribution of the Deutsche Forschungsgemeinschaft (DFG) supported project "Sonderforschungsbereich 754: Climate-Biogeochemistry Interactions in the Tropical Ocean" (http://www.sfb754.de). The WOA13 data are available at https://www.nodc.noaa.gov/OC5/woa13/woa13data.html. The 1/12° global HYCOM+NCODA Ocean Reanalysis was funded by the U.S. Navy and the Modelling and Simulation Coordination Office. Computer time was made available by the
DoD High Performance Computing Modernization Program. The output is publicly available at http://HYCOM.org.

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




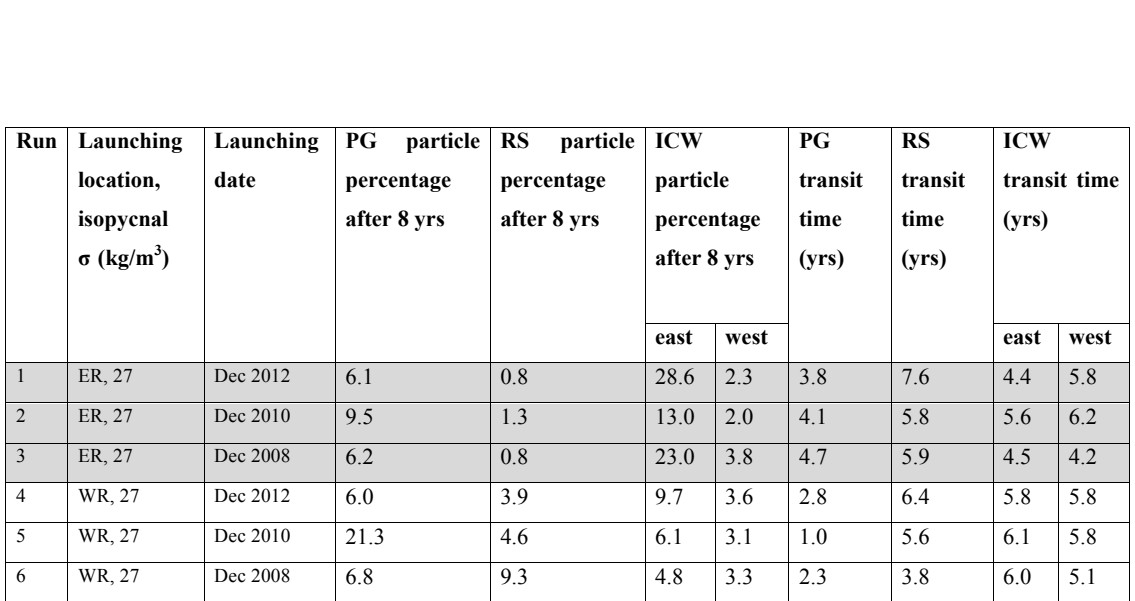

| Run | Launching location, isopycnal σ (kg/m³) | Launching date | PG particle percentage after 8 yrs | RS particle percentage after 8 yrs | ICW particle percentage after 8 yrs | | PG transit time (yrs) | RS transit time (yrs) | ICW transit time (yrs) | |
|---|---|---|---|---|---|---|---|---|---|---|
| | | | | | east | west | | | east | west |
| 1 | ER, 27 | Dec 2012 | 6.1 | 0.8 | 28.6 | 2.3 | 3.8 | 7.6 | 4.4 | 5.8 |
| 2 | ER, 27 | Dec 2010 | 9.5 | 1.3 | 13.0 | 2.0 | 4.1 | 5.8 | 5.6 | 6.2 |
| 3 | ER, 27 | Dec 2008 | 6.2 | 0.8 | 23.0 | 3.8 | 4.7 | 5.9 | 4.5 | 4.2 |
| 4 | WR, 27 | Dec 2012 | 6.0 | 3.9 | 9.7 | 3.6 | 2.8 | 6.4 | 5.8 | 5.8 |
| 5 | WR, 27 | Dec 2010 | 21.3 | 4.6 | 6.1 | 3.1 | 1.0 | 5.6 | 6.1 | 5.8 |
| 6 | WR, 27 | Dec 2008 | 6.8 | 9.3 | 4.8 | 3.3 | 2.3 | 3.8 | 6.0 | 5.1 |
| 7 | ER, 26.4 | | 18.6 | 4.1 | 13.0 | 4.5 | 2.9 | 5.4 | 5.2 | 5.6 |
| 8 | WR, 26.4 | | 2.3 | 17.6 | 12.2 | 18.2 | 1.1 | 1.6 | 5.2 | 3.3 |
| 9 | ER, 27.4 | | 8.2 | 1.3 | 6.1 | 3.6 | 5.2 | 6.4 | 5.3 | 6.1 |
| 10 | WR, 27.4 | | 9.7 | 5.2 | 2.6 | 4.2 | 4.9 | 6.3 | 6.1 | 6.0 |

Table 1: Trajectory calculations of percentages and transit times of advective Lagrangian particles for different runs performed with a length of 8 years on different isopycnals (σ = 26.4 kg/m³; 27 kg/m³; 27.4 kg/m³) for this study. Transit times are defined by the times where 50% and 75% of the particles crossed the section from the three major source regions (PG, RS, and southern Indian Ocean) to the two release areas (ER and WR). Percentages refer

10 to the total number of Lagrangian particles that have crossed the section after the whole time span of the simulation (8 years). Release locations are defined as circles with a radius of twice the grid spacing, thus 1/6°, around the coordinates of the release points (abbreviations defined in section 2.2). The number of released floats is 10000 for all runs that are listed here.

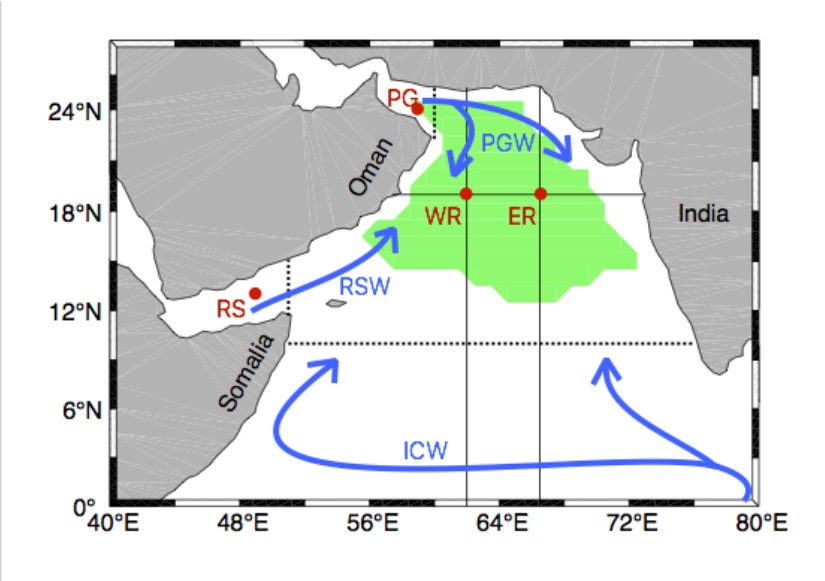

**Figure 1: The green patch sketches the location of the Arabian Sea Oxygen Minimum Zone (ASOMZ) defined by oxygen of less than 10 μmol/kg. Schematic pathways of the three major intermediate source water masses in the northwest Indian Ocean are marked in blue: Indian Central Water (ICW), Red Sea Water (RSW), and Persian Gulf Water (PGW). Location of four particle release points (western basin (WR), eastern basin (ER), Persian Gulf (PG), Red Sea (RS)) are shown as red dots. Black solid lines indicate the sections shown in Fig. 3. Sections, that need to be crossed by the Lagrangian particles to define the source regions for the ICW, RSW, and PGW are marked as black dashed lines.**





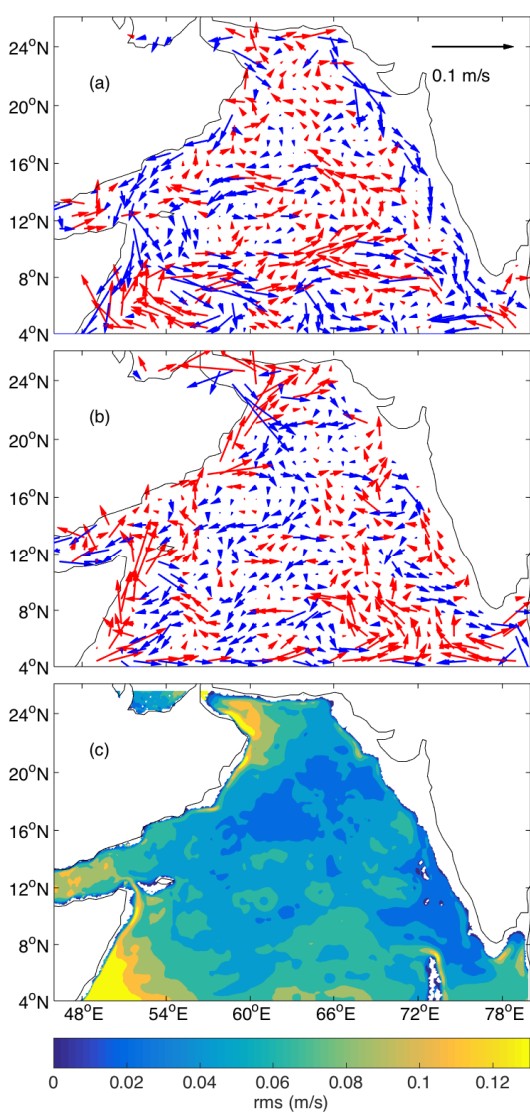

Figure 2: **Mean seasonal velocity for the Arabian Sea based on HYCOM data on the isopycnal surface of σ=27 kg/m³ for (a) northeast (November-February) and (b) southwest monsoon (June-September) averaged for 2000-2012. The velocity field is spatially filtered (0.6° x 0.6° window) and presented on a grid with the same resolution. Northward (southward) directed currents are shown in red (blue). (c) Root Mean Square (rms) error of the annual mean velocities from 2000-2012.**





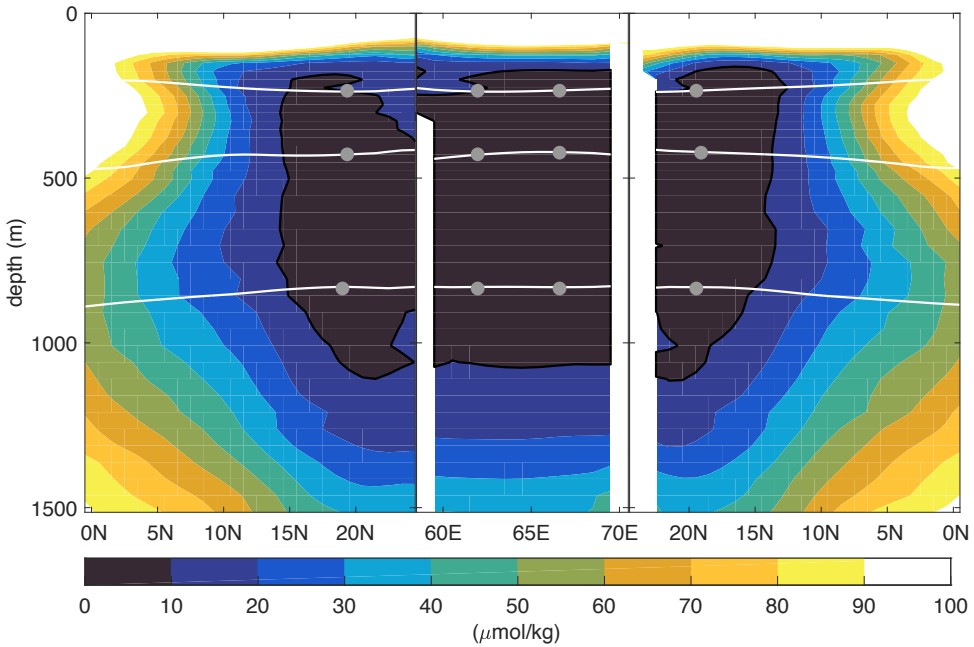

**Figure 3: Annual mean of dissolved oxygen concentration along 62°E (left), 66.5° E (right) and 19° N (middle) from the WOA 13 climatology (see Figure 1). Advective pathways from Lagrangian particles are calculated on three isopycnals (26.4, 27, 27.4 kg/m³)**
5   **shown as white lines. The grey dots mark the release points in the western (WR) and eastern (ER) basin on each isopycnal level.**

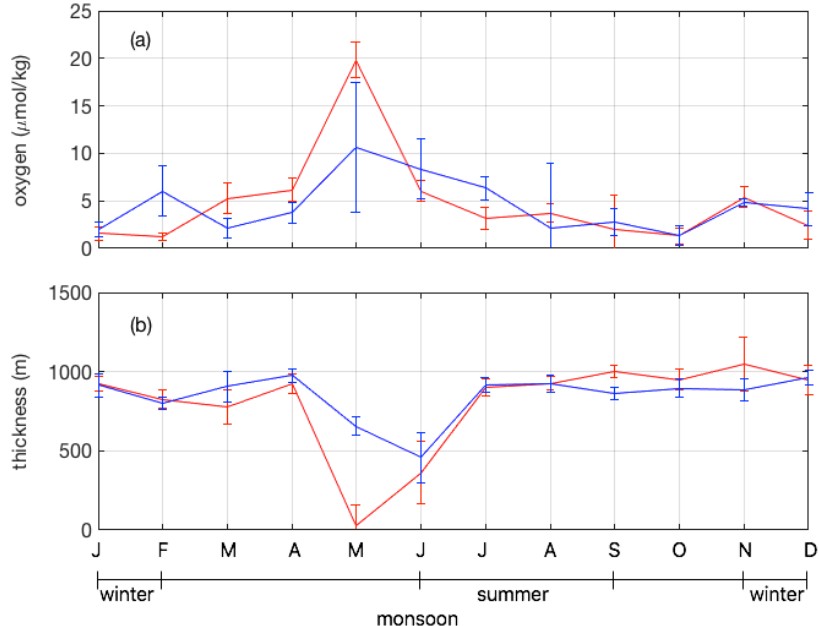

**Figure 4: (a) Mean seasonal cycle of dissolved oxygen concentration on the isopycnal surface of σ=27 kg/m³ at the location of the ER (red) and the WR point (blue) from observations. (b) Mean seasonal cycle of the thickness of the layer containing oxygen of less than 10 μmol/kg based on WOA 13 at the location of the ER (red) and the WR point (blue). The error bars show the spatial standard deviation in an area of 2° x 2° centered around the release point.**





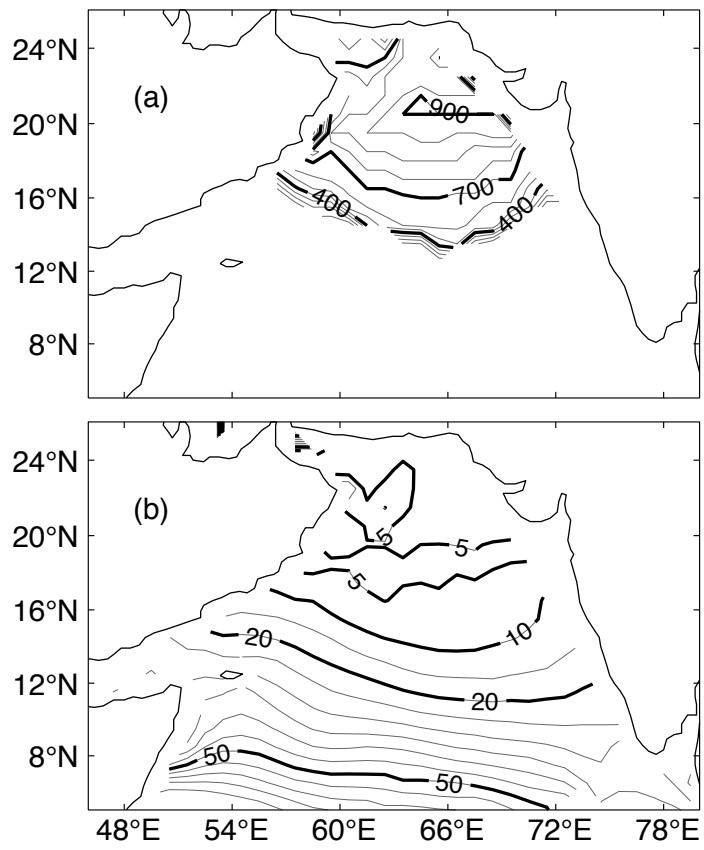

Figure 5: (a) Thickness (in m) of the layer containing oxygen of less than 10 µmol/kg based on climatological data from WOA 13. (b) Oxygen concentration (in µmol/kg) on the $\sigma = 27$ kg/m$^3$ isopycnal of WOA 13.



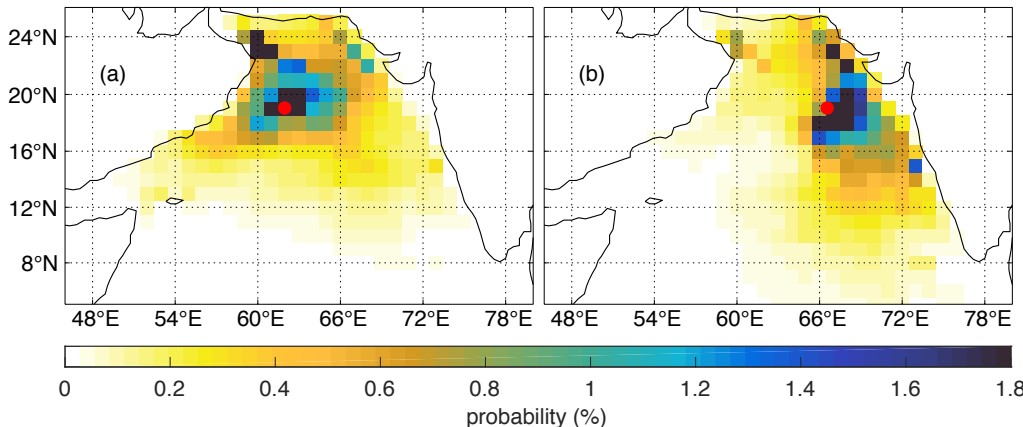

**Figure 6: Probability that a 1°x1° bin is occupied by a Lagrangian particle during the time span of 8 years for backward trajectory calculations from (a) the western (WR) and (b) the eastern (ER) part of the ASOMZ along the isopycnal σ=27 kg/m³. Red dots mark the location of the WR and ER points.**



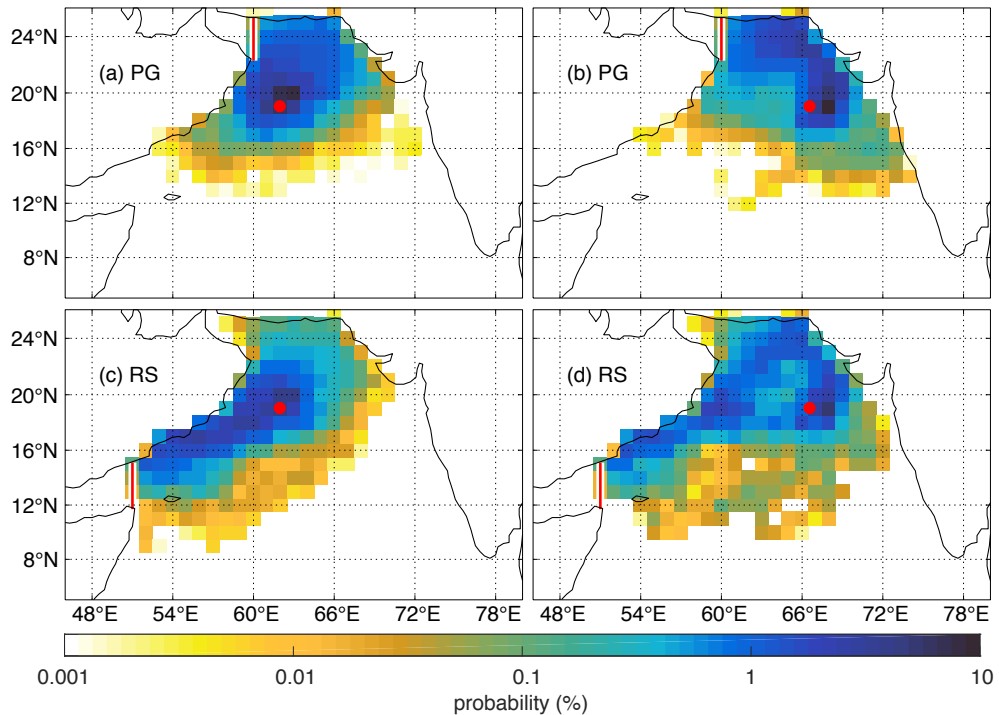

**Figure 7: Lagrangian particle position probability maps show the most pronounced pathways of fluid particles along the isopycnal σ=27 kg/m³ for the backward trajectory analysis, entering the Persian Gulf from (a) WR and (b) ER and the Red Sea from (c) WR and (d) ER. Eastern (ER) and western release (WR) locations in the OMZ are marked in red, as well as the sections, that need to be crossed to define the source regions.**




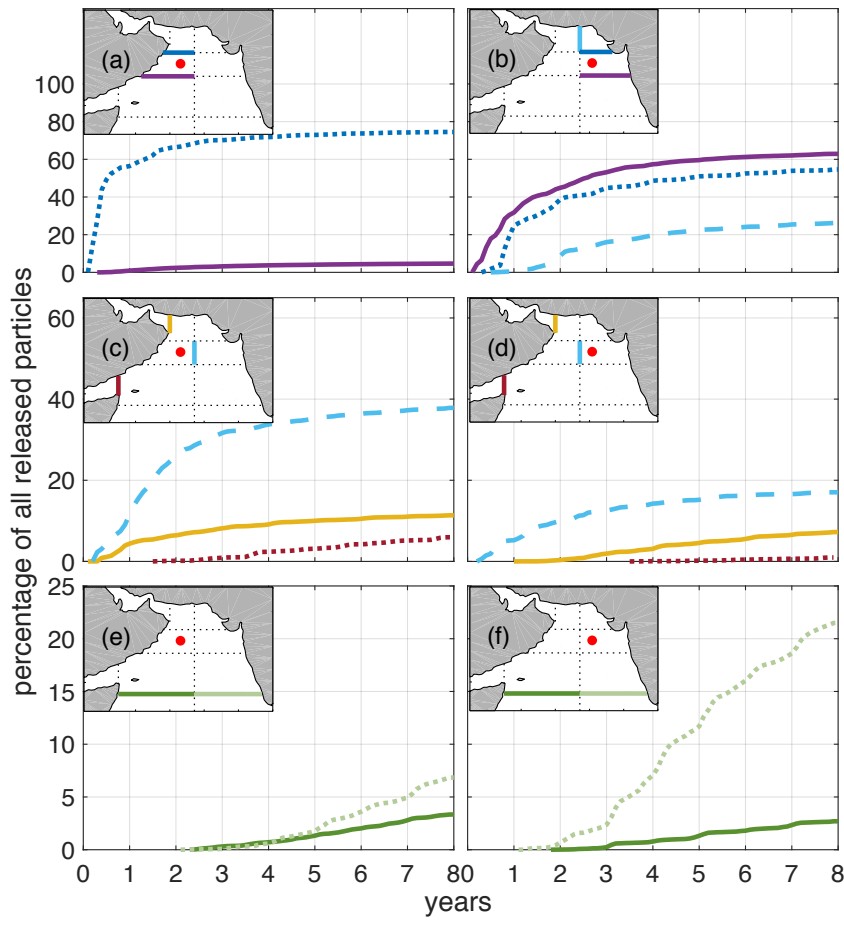

Figure 8: Cumulative point to point transit times of Lagrangian particles calculated between distinct sections (see maps) along their main pathways and their release points in the western basin (left column) and eastern basin (right column), respectively. Sections are along 21° N shown as dotted dark blue line and 17° N as solid purple line (a, b), 64.3° E as dashed light blue line (b, c, d), 60° E as solid yellow line, 51° E as dotted red line (c, d), 10° N as dotted light green line (east) and solid green line (west) (e, d). The red dots mark the launching position of the backward trajectories. See section 2.2 for a detailed description.



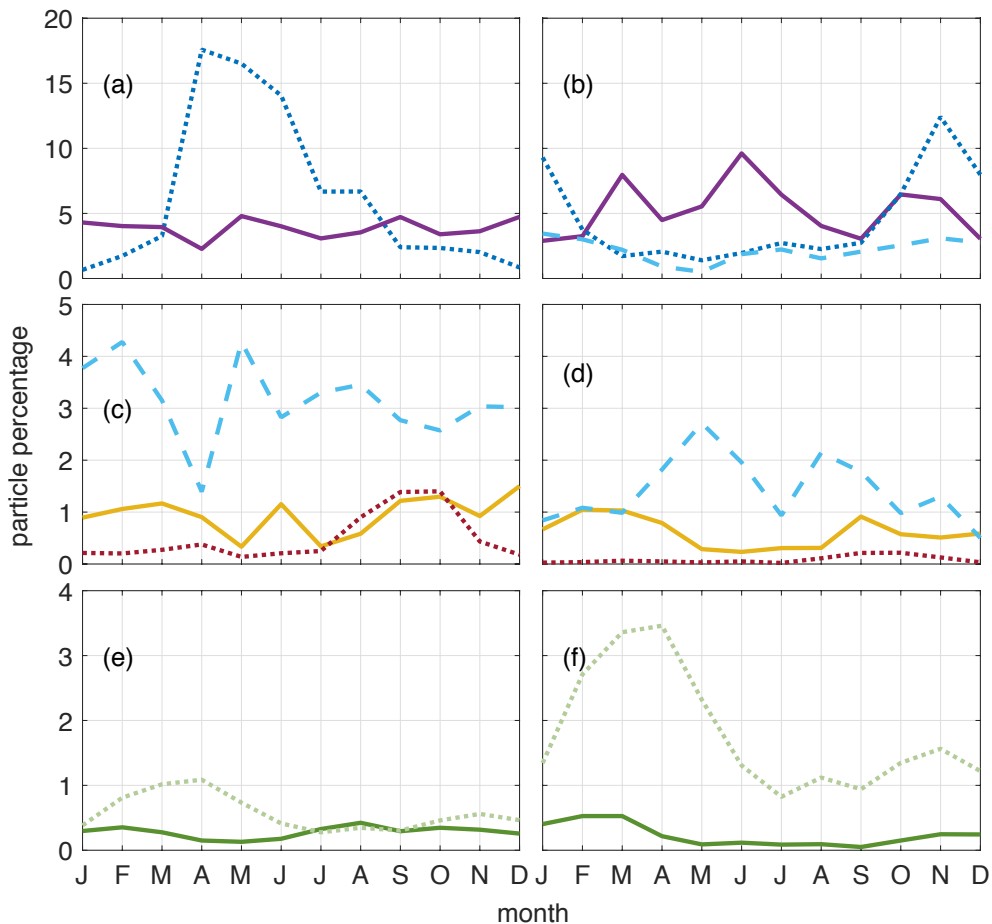

**Figure 9: Mean seasonal cycle of particle percentage travelling between distinct sections along their main pathways and their release points in the western basin (left column) and the eastern basin (right column), respectively. Sections are along 21° N shown as dotted dark blue line and 17° N as solid purple line (a, b), 64.3° E as dashed light blue line (b, c, d), 60° E as solid yellow line, 51° E as dotted red line (c, d), 10° N as dotted light green line (east) and solid green line (west) (e, d). For line colour and type please see also Figure 8.**