# Peer review of "Seasonal variability of the Arabian Sea intermediate circulation and its impact on seasonal changes of the upper Oxygen Minimum Zone"

_Ocean Science, 2020_

## Referee Comment (RC1) · Anonymous Referee #1 · 26 Feb 2020

This study uses a Lagrangian approach to better understand the pathways into the Arabian Sea OMZ using HYCOM velocity fields. This is an extremely important topic and the results have the potential to be very valuable for the millions of people that depend on fish catches in the Arabian Sea as their primary form of sustenance, among others. The analysis has potential to produce a useful contribution to the current literature and has very clearly stated objectives, but I have many concerns, listed below. I cannot recommend this article for publication in its current form, but would be willing to review another version pending major revisions.

Some temporal discrepancies: The WOA13 monthly dissolved oxygen climatology cov-

ers 1955-2012, the HYCOM velocities are daily from 2000-2012, and the NCEP CFSR forcing is used from 1995-2012. Are the years in the actual analysis consistent? How can this version of HYCOM have any forcing from 1995-2000 if it starts in 2000? NCEP CFSR only exists until March 2011, at which point most systems switch their forcing to CFSv2. The authors need to make section 2.1 more clear.

A major issue I have is that HYCOM is not a biophysical model. A comparison with a biophyscial model might be more appropriate. Comparison between model currents and real dissolved oxygen measurements is not necessarily realistic. Also, the temporal sampling between currents and dissolved oxygen should be the same. I understand that there is a limited number of dissolved oxygen observations, in which case climatological currents should be used as well.

The authors validate their HYCOM velocity data with YoMaHa'07, which is based on observational Argo data. Why use HYCOM at all if the authors have observational YoMaHa currents that they can compare with the observation-based climatology of WOA13? It does not make sense to compare model currents to explain observations when observational currents are available, especially if the authors are using a Lagrangian approach. Why not use observed Lagrangian data, e.g. drifters? There is plenty of drifter data in this region (see various papers by R. Lumpkin).

YoMaHa currents in figure S1 are from 1997-2007 and HYCOM is implied to be from 2000-2012 based on the text. If they are not the same time period, please correct this.

There is no discussion of non-advective processes that influence changes in oxygen. A full budget analysis might be outside of the scope of this paper, but there is sufficient velocity data to quantify influence of upwelling on biological productivity, particularly because it drives a lot of the biophysical dynamics in this region, particularly during the summer monsoon season and should not be ignored.

Page 2: "the south eastern parts of the tropical ocean poor ventilation south of the subtropical gyre circulation (Luyten et al., 1983)" Check grammar. Also, how is the

tropical ocean south of the subtropical gyre?

Page 4, line 12: "it's" to "its"

Page 6, line 1: I'm not sure why this is a good thing. Vertical advection is an important part of the flow in this region and would be realistic.

First paragraph of page 6: There are a lot of assumptions being made and not many sources. I am not convinced that these are realistic assumptions.

Figure 1: Is this schematic applicable year-round? I imagine the circulation pathways would be very different during the summer and winter monsoon.

With all the discussion of calculating pathways and trajectories, it would be nice to have a figure explicitly showing some of this, as is promised in the beginning of section 3.2, rather than only particle position probability maps. Many of the conclusions that the authors are making are very difficult to obtain from the figures.

Page 9: Why discuss Figure 2 after Figure 4? Throughout the paper there is an unusual amount of jumping between figures, giving the impression that it is poorly organized.

Section 3.1: None of this seems new, especially the second half. Monsoon circulation has been well-studied for decades.

The entire results section is very hard to read, partly due to the writing and partly due to the lack of clarity of the figures. I recommend that the authors read some papers on studies that use drifters for reference to show clear oceanic pathways. A good example is: Drouin, K. L., & Lozier, M. S. (2019). The surface pathways of the South Atlantic: Revisiting the cold and warm water routes using observational data. Journal of Geophysical Research: Oceans, 124(10), 7082-7103.

Add references: Lachkar, Z., Lévy, M., & Smith, K. S. (2019). Strong intensification of the Arabian Sea oxygen minimum zone in response to Arabian Gulf warming. Geophysical Research Letters, 46(10), 5420-5429.

Shenoy, D. M., Suresh, I., Uskaikar, H., Kurian, S., Vidya, P. J., Shirodkar, G., ... & Naqvi, S. W. A. (2020). Variability of dissolved oxygen in the Arabian Sea Oxygen Minimum Zone and its driving mechanisms. Journal of Marine Systems, 103310.
* * *

---

## Referee Comment (RC2) · Anonymous Referee #2 · 8 Mar 2020

General Comments

The manuscript addresses important concerns relating to the oxygen minimum zone in the Arabian Sea, but could benefit from a more complete comparison of the model output to observational data, including acknowledgement of existing observational publications.

Specific comments

It is not clear to me how "intermediate" depth is defined. On Page 2 Line 7 the OMZ is stated as 200-700 m depth, and Fig 3 has particle paths spanning from ∼200-800 m depth, so I assume this must be the range, but it should be stated more clearly in the

text (maybe in parenthesis behind the first reference to "intermediate depth)

Page 4 Lines 26-29 not necessary

Page 5 line 21+ say model output were validated using YoMaHa data, but no detail is provided other than stating it "agrees very well."

The manuscript states a lack of observational data, but several circulation studies have been done in this region that are not acknowledged in this manuscript. The manuscript would benefit from a more detailed description of how model output compare to both YoMaHa observations as well as previously published observations. Suggestions here: • Zhankun Wang et al 2014 (Deep Sea Research) Seasonal and annual variability of vertically migrating scattering layers in the northern Arabian Sea • Zhankun Wang et al 2013 (Deep Sea Research) High salinity events in the northern Arabian Sea and Sea of Oman • Sarah Stryker Vitale et al 2017 (Dynamics of Atmospheres and Oceans) Circulation analysis in the northwest Indian Ocean using ARGO floats and surface drifter observations, and SODA reanalysis output

Page 9 Line 17- state approximate depths of isopycnals (here and elsewhere) in the text so the reader does not have to refer to the figure throughout

The last sentence of the conclusion is not clearly written.

Technical Comments

Page 4 line 12 "it's" should be "its"

Page 8 Line 25- Not clear to me what "lack of 3 month" means- typo?

Page 14 Line 27- "weak" misspelled

---

## Author Comment (AC1) · 29 Apr 2020

Anonymous Referee #1 Received and published: 26 February 2020

This study uses a Lagrangian approach to better understand the pathways into the Arabian Sea OMZ using HYCOM velocity fields. This is an extremely important topic and the results have the potential to be very valuable for the millions of people that depend

on fish catches in the Arabian Sea as their primary form of sustenance, among others. The analysis has potential to produce a useful contribution to the current literature and has very clearly stated objectives, but I have many concerns, listed below. I cannot recommend this article for publication in its current form, but would be willing to review another version pending major revisions.

Reply to reviewer #1 We would like to thank the reviewer for taking the time and for providing constructive and very specific comments, which helped to improve the manuscript considerably. We rearranged the order of figures and included two completely new figures (Figs. 6 and 11). We also rewrote some parts of the manuscript to make the results and the discussion easier to follow. We have carefully addressed his/her comments. The point-by-point responses follow below the specific comments.

1) Some temporal discrepancies: The WOA13 monthly dissolved oxygen climatology covers 1955-2012, the HYCOM velocities are daily from 2000-2012, and the NCEP CFSR forcing is used from 1995-2012. Are the years in the actual analysis consistent? How can this version of HYCOM have any forcing from 1995-2000 if it starts in 2000? NCEP CFSR only exists until March 2011, at which point most systems switch their forcing to CFSv2. The authors need to make section 2.1 more clear.

It is right that the WOA 13 monthly dissolved oxygen climatology and the HYCOM velocities cover different time periods, but this is not relevant for the analysis of the Lagrangian pathways. In this study WOA 13 oxygen data are exclusively used to motivate the setup of the experiment. An overview about location, spatial extension (Fig. 3) and seasonal cycle (Fig. 4) of the ASOMZ is given in order to define e.g. the locations as well as the isopycnal of the release of the particles. The actual analysis of the seasonal variability of the circulation is done only with the HYCOM velocities and is not relying on the time covered by the oxygen observations. Please see text: "The global observational dissolved oxygen climatology of the World Ocean Atlas 2013 (WOA13), is used to plan and setup the study. The monthly mean data cover a period from 1955-2012 and are available with a spatial resolution of 1° x 1° interpolated on

OSD
102 depth levels (Garcia et al., 2013)." The HYCOM reanalysis data are available from August 1995 until December 2012. Due to some gaps in the first years of the analysis, we decided only to use the data from 2000 onward for the trajectory analysis. Thus the forcing is actually from 1995 until 2012. We stated that point more clearly in the script now: "Due to missing data in the early years of the reanalysis, the model velocity output used here for the trajectory analysis were daily snapshots for the time period from January 2000 to December 2012." We also included the correct forcing for the later years in the text: "The surface forcing is from the National Center for Environmental Prediction (NCEP) Climate Forecast System Reanalysis (CFSR, 1995-2011) and Climate Forecast System Version 2 (CFSv2, 2011-2012)."

2) A major issue I have is that HYCOM is not a biophysical model. A comparison with a biophyscial model might be more appropriate. Comparison between model currents and real dissolved oxygen measurements is not necessarily realistic. Also, the temporal sampling between currents and dissolved oxygen should be the same. I understand that there is a limited number of dissolved oxygen observations, in which case climatological currents should be used as well.

Thanks for bringing this point into the discussion. We spend some time planning the experiment and ended up with the decision not to use a biophysical model, because there is still some discussion on the dynamics of the OMZ in the Arabian Sea as well as biophysical models reveal large uncertainties in comparison to observations such as they fail to reproduce the OMZ. Based on studies on the equatorial Pacific from global coupled biogeochemical circulation models Dietze and Löptien (2013) point out that poor model performance with respect to oxygen minimum zones is related to a deficient representation of ventilation pathways rather than being associated with a deficient representation of biogeochemical processes (i.e. respiration). Therefore we put our focus on the physical processes and chose the HYCOM model, which has a high "spatial resolution of 1/12° in longitude and latitude with 40 depth levels" and "a realistic bathymetry based on the General Bathymetric Chart of the Oceans (GEBCO) and uses
isopycnal coordinates in the open, stratified ocean, changes to terrain-following coordinates in shallower and coastal regions and uses z-level coordinates in the mixed layer." "The variable vertical coordinates are beneficial for the model to better reproduce the circulation near out-/overflow regions compared to typical z-level models, which would generally have problems to resolve the shallower coastal regions properly (see also Bleck and Boudra, 1981; Bleck and Benjamin, 1993; Bleck, 2002). The latter is important for the analysis of the supply of Persian Gulf Water (PGW) and Red Sea Water (RSW) through the Gulf of Oman and the Gulf of Aden respectively." As already mentioned in the manuscript, we included climatological current and compared them with the ones calculated from the velocity time series. ("To detect the interannual variability the runs with the duration of 13 years used for the statistics were compared to a climatological run, which was performed with velocity fields with the mean daily velocity of the 13 years at each grid point and day.") The differences that are found are stated in section 3.5. Coming to the temporal sampling of the particles and trajectories, the statements we make about the seasonal cycle refer to climatological monthly crossings over the selected sections (Fig. 10). Comparing this with the monthly oxygen climatology, the temporal sampling is the same, even if the input comes from different time scales. So we think that it is appropriate to draw conclusions from that.

3) The authors validate their HYCOM velocity data with YoMaHa'07, which is based on observational Argo data. Why use HYCOM at all if the authors have observational YoMaHa currents that they can compare with the observation-based climatology of WOA13? It does not make sense to compare model currents to explain observations when observational currents are available, especially if the authors are using a Lagrangian approach. Why not use observed Lagrangian data, e.g. drifters? There is plenty of drifter data in this region (see various papers by R. Lumpkin).

Ventilation in the OMZ layers of the AS is mainly facilitated by three major source water masses (Indian Central Water, Persian Gulf Water and Red Sea Water) circulating at intermediate depth. Therefore, the isopycnal layers of the Lagrangian analysis
performed for our study are found at intermediate depth between 250 to 900 m. Unfortunately, in this depth range observational data are sparse and thus we have to rely on model data. Due to the lack of observational velocity data at intermediate depth, the validation of the model is done with observations at the surface, where the data coverage is sufficient with e.g. drifter or Argo data. The YoMaHa'07 climatology based on Argo observational data only provides velocity data at the surface and at 1000 m depth. This is also stated in the Introduction: "While the circulation of the upper-ocean is fairly well known from drifter data (Shenoi et al., 1999) and satellite altimetry (Beal et al., 2013) precise subsurface ventilation pathways of water masses entering the AS beneath the surface layer are less well understood in detail due to a lack of observational data (McCreary et al., 2013)..." as well as in chapter 2.1: "For a validation of the HYCOM velocity data we compared the near-surface circulation of HYCOM with the climatology of YoMaHa'07, which is based on observational data obtained from Array of Real-time Geostrophic Oceanography (ARGO) floats (Lebedev et al., 2007). This choice is motivated by the lack of observational data at intermediate depths between 200 and 800 m in the Arabian Sea."

4) YoMaHa currents in figure S1 are from 1997-2007 and HYCOM is implied to be from 2000-2012 based on the text. If they are not the same time period, please correct this.

Thank you for pointing this out. We changed the time period from 1997 - 2007 for both data sets now for the validation of the model and updated figure S1. We modified the text: "For a validation of the HYCOM velocity data we compared the near-surface circulation of HYCOM with the climatology of YoMaHa'07, which is based on observational data obtained from Array of Real-time Geostrophic Oceanography (ARGO) floats (Lebedev et al., 2007). This choice is motivated by the lack of observational data at intermediate depths between 200 and 800 m in the Arabian Sea. YoMaHa'07 provides a  $1^{\circ}x1^{\circ}$  bin averaged monthly climatology of the surface velocity for the time period from 1997 to 2007. Therefore HYCOM data used for the validation were monthly averaged for the time period from 1997 to 2007 and  $1^{\circ}x1^{\circ}$  bin averaged in accordance with the
YoMaHa'07 climatology."

5) There is no discussion of non-advective processes that influence changes in oxygen. A full budget analysis might be outside of the scope of this paper, but there is sufficient velocity data to quantify influence of upwelling on biological productivity, particularly because it drives a lot of the biophysical dynamics in this region, particularly during the summer monsoon season and should not be ignored.

The influence of upwelling on biological productivity and hence for the advection of the ASOMZ is discussed now in more detail. In addition, we included a new figure (Fig. 11), which schematically shows main pathways and areas of net primary production for the monsoon and intermonsoon seasons to comprehend the results and the impact of the seasonal biological production. Please, see text: Discussion: " During the summer monsoon enhanced upwelling occurs along the western boundary leading to the incidence of phytoplankton blooms, resulting in a region of highest biological productivity in the world ocean (Qasim, 1982). " "Lagrangian particles cross regions with high primary production during the long-distance advection while looping around the northern part of the basin. A study of seasonal vertically migrating scattering layers reveals a rapid increase of biomass in the northern Arabian Sea in the layer between 250 and 450 m depth during the period of June to November (Wang et al., 2014). Resulting high consumption rates in the outflow region of the Gulf of Oman in winter (Fig. 11, Acharya and Panigrahi, 2016; Lachkar et al., 2018) might lead to a transport of lower oxygenated water, which might be responsible for the minimum of oxygen concentration in the eastern basin in winter (Fig. 4a)." "Compared to the strong variability of the western boundary current, the seasonal oxygen cycle in the western basin is weak (Fig. 4a). This could be explained by the loss of oxygen via consumption when RSW passes the area of strong primary production off the coast of Oman (Acharya and Panigrahi, 2016) during the summer monsoon (Fig 11)."

6) Page 2: "the south eastern parts of the tropical ocean poor ventilation south of the subtropical gyre circulation (Luyten et al., 1983)" Check grammar. Also, how is the
tropical ocean south of the subtropical gyre?

Thank you for noticing that. We changed the sentence to: "Regions of sluggish ventilation can be found in all ocean basins e.g. in the eastern parts of the tropical Pacific and Atlantic Ocean southeast/northeast of the subtropical gyre circulation (Luyten et al., 1983) and in the northern Indian Ocean due to the lack of ventilation from the north. Weak ventilation combined with high biological production in upwelling regions and thus enhanced oxygen consumption by sinking organic matter results consequently in very low levels of dissolved oxygen below the surface (Stramma et al., 2008; Gilly et al., 2013)."

7) Page 4, line 12: "it's" to "its" We changed that.

8) Page 6, line 1: I'm not sure why this is a good thing. Vertical advection is an important part of the flow in this region and would be realistic.

You are absolutely right that vertical advection, especially in the upwelling regions is an important part of the flow. However, Banse et al. (2014) considers the supply of oxygen to the ASOMZ to be mainly along isopycnals. As far as we know this is the state of the art knowledge as also other recent studies refer to it (Shenoy et al., 2020). Thus we assume that the assumptions are sufficient for the scope of our paper focusing on the advection in the AS at intermediate depths. We state in the paper: "For the AS, the supply of oxygen was suggested by Banse et al. (2014) to be on the isopycnal surfaces of 27 kg/m3 associated with depths between 300-500 m."

9) First paragraph of page 6: There are a lot of assumptions being made and not many sources. I am not convinced that these are realistic assumptions.

We hope that we could motivate our assumptions more clearly by indicating relevant references and figures. "Our experiments are based on the assumption that PGW and RSW are the main local source water masses that are relevant for the ventilation of the ASOMZ (Prasad et al., 2001; Beal et al., 2000; Shankar et al., 2005) and that

OSD
the oxygen rich waters follow largely their isopycnal layer horizontally. The annual mean of dissolved oxygen from WOA 13 climatology is shown in Figure 3 to give an overview about location and spatial extension of the ASOM as well as the isopycnal layers associated with the density of the PGW and RSW."

10) Figure 1: Is this schematic applicable year-round? I imagine the circulation pathways would be very different during the summer and winter monsoon.

Yes, you are right. The pathways are changing during the different monsoon seasons as described in the manuscript. The schematic (Fig. 1) aims to give a rough overview about the main water masses at intermediate depths as well as their main pathways. Additionally, we included a new schematic showing the main advective pathways at intermediate depths for the different monsoon seasons (Fig. 11) which helps to point out the results of our study.

11) With all the discussion of calculating pathways and trajectories, it would be nice to have a figure explicitly showing some of this, as is promised in the beginning of section 3.2, rather than only particle position probability maps. Many of the conclusions that the authors are making are very difficult to obtain from the figures.

We have now included two new figures that hopefully help to comprehend the results pointing out in the manuscript. The new figure 6 shows a random selection of the trajectories connecting the eastern and western release point with the Red Sea and the Persian Gulf. Please see text in section 3.2 right at the beginning:" Figure 6 shows exemplary trajectories connecting the release locations with the marginal seas. This subsample of advective Lagrangian pathways already shows that the majority of particles follows distinct pathways." The new figure 11 schematically shows the main advective pathways at intermediate depths for different monsoon seasons wrapping up the content of figures 8 to 10 (new figure numbers) and we hope that this helps to follow our conclusions.

12) Page 9: Why discuss Figure 2 after Figure 4? Throughout the paper there is

OSD
an unusual amount of jumping between figures, giving the impression that it is poorly organized.

We are sorry that the order of the figures confused you. As we refer to some of them already in section 2, we sorted them by their first appearance in the manuscript. However, we took your comment in consideration and reorganised the figures in a way that they now appear in order of their main discussion. We are aware of the fact that we refer quite often to figures that have been discussed earlier in chapter 2. Data and method, especially figures 2 to 5. We considered it to be helpful and not distracting for the reader to refer to them more often as these are fundamental figures on which the experimental setup is based. In the discussion we refer more often to the new figure (Fig. 11), which schematically show the results and prevents from jumping between the figures as well.

13) Section 3.1: None of this seems new, especially the second half. Monsoon circulation has been well-studied for decades.

It is right that near-surface monsoon circulation has been studied before but there is still a lack of observational velocity data at intermediate depth. In this study, we describe the circulation at intermediate depth, referring to the HYCOM velocities at the isopycnal surface of 27 kg/m3 (depth  $\sim$  450 -500 m). We think it is essential to show the velocities on which we base the trajectory analysis and helpful for the reader as we refer to it later in the text as well. Thus we would like to keep it in there.

14) The entire results section is very hard to read, partly due to the writing and partly due to the lack of clarity of the figures. I recommend that the authors read some papers on studies that use drifters for reference to show clear oceanic pathways. A good example is: Drouin, K. L., & Lozier, M. S. (2019). The surface pathways of the South Atlantic: Revisiting the cold and warm water routes using observational data. Journal of Geophysical Research: Oceans, 124(10), 7082-7103.

Thank you for the recommended paper. We hope that the new figures (Figs. 6 and 11)
help the reader to follow our thoughts. The new figure 6 shows a random selection of the trajectories connecting the eastern and western release point with the Red Sea and the Persian Gulf. The new figure 11 schematically shows the main advective pathways at intermediate depths for different monsoon seasons. We are aware that Lagrangian analysis offers a lot of different methods to study and to present ocean circulation. It turned out that probability maps of the Lagrangian particle position as well as the point to point transit time analysis seemed to be the best methods to visualize the trajectories calculated in this study. These methods are appropriate to other studies on Lagrangian trajectories as e.g. in Gary et al., 2011 and in van Sebille et al., 2018 showing a review about "Lagrangian ocean analysis: Fundamentals and practices".

15) Add references: Lachkar, Z., Lévy, M., & Smith, K. S. (2019). Strong intensification of the Arabian Sea oxygen minimum zone in response to Arabian Gulf warming. Geophysical Research Letters, 46(10), 5420-5429. Shenoy, D. M., Suresh, I., Uskaikar, H., Kurian, S., Vidya, P. J., Shirodkar, G., ... & Naqvi, S. W. A. (2020). Variability of dissolved oxygen in the Arabian Sea Oxygen Minimum Zone and its driving mechanisms. Journal of Marine Systems, 103310. We added the refernces to the text: "That this season is crucial for the ventilation of the OMZ with PGW was shown by Lachkar et al. (2019) with a model sensitivity study." "Oxygenated Indian Central Water (ICW) enters the AS at intermediate depth (200 – 500 m; Shenoy et al., 2020) from the south (Fig. 1)." The northward advection into the eastern basin along the Indian coast shows a second maximum during winter monsoon (Fig. 11a), which can be confirmed by observations showing a supply of oxygenated ICW during that time (Shenoy et al., 2020).
Fig. 1. Figure 3: Annual mean of dissolved oxygen concentration along 62°E (left), 66.5° E (right) and 19° N (middle) from the WOA 13 climatology (see Figure 1). Advective pathways from Lagrangian particles a

0 26.4 27.0 500 depth (m) 27.4 1000 1500 70E 5N 0N 5N 10N 15N 20N 60E 65E 20N 15N 10N 40 50 60 70 10 20 30 80 90 0

---

## Author Comment (AC2) · 29 Apr 2020

General Comments The manuscript addresses important concerns relating to the oxygen minimum zone in the Arabian Sea, but could benefit from a more complete comparison of the model output to observational data, including acknowledgement of existing

observational publications.

Reply to reviewer #2 We would like to thank the reviewer for taking the time and for providing constructive and very specific comments, which helped to improve the manuscript considerably. We rearranged the order of figures and included two completely new figures (Figs. 6 and 11). We also rewrote some parts of the manuscript to make the results and the discussion easier to follow. We have carefully addressed his/her comments. The point-by-point responses to the specific comments follow below.

Specific comments 1) It is not clear to me how "intermediate" depth is defined. On Page 2 Line 7 the OMZ is stated as 200-700 m depth, and Fig 3 has particle paths spanning from 200-800 m depth, so I assume this must be the range, but it should be stated more clearly in the text (maybe in parenthesis behind the first reference to "intermediate depth)

This is a good point. We included the definition of "intermediate" depth now more clearly in the manuscript. Abstract: ". . . to investigate the advective pathways of Lagrangian particles into the Arabian Sea OMZ at intermediate depths between 200 and 800 m." Section 2.2: ". . .OMZ by performing backward trajectories and to draw inferences of the basin wide spread of oxygen at intermediate depth (200 – 800 m)."

2) Page 4 Lines 26-29 not necessary

We would like to keep the paragraph. It might not be necessary but we consider it as helpful for the reader.

3) Page 5 line 21+ say model output were validated using YoMaHa data, but no detail is provided other than stating it "agrees very well."

We added a carefully description of the near-surface circulation obtained from YoMaHA07 as well as HYCOM. Additionally, we refer to the analysis of near-surface circulation in the northwestern Indian Ocean based on drifter data by Vitale et al. (2017).

"The complex circulation pattern at the near-surface, which is strongly affected by the seasonal Asian monsoon (Schott et al., 2009) is well described by the HYCOM data reflecting all (reversing) currents that are relevant for the AS. During the winter monsoon, the Somali Current (SC) flows southwestward along the coast of Somalia (Fig. S1a, b). The Northeast Monsoon Current (NMC) sets westward at the southern tip of India and supplies the West Indian Coast Current (WICC), which is flowing northward along the coast of India (Figs. 1a, b). During the summer monsoon, the Somali Current (SC) and the Ras al Hadd Jet (RHJ; also called East Arabian Current (Vitale et al., 2017)) flow northeastward along the coast of Somalia and Oman (Fig. S1c, d). A strong gyre, the Great Whirl (GW), can be identified, which generally develops off the coast of Somalia during the summer monsoon season (Figs. S1c, d). In accordance to observations, the West Indian Coast Current (WICC) flows southward along the coast of India and feeds the Southwest Monsoon Current (Figs. 1c, d). The comparison of the near-surface circulation obtained from HYCOM and ARGO agrees very well during the winter and summer monsoon (see supplement Fig. S1). Additionally, an analysis of seasonal surface velocities in the AS (Vitale et al., 2017; their Figs. 2a, 3a), which is based on a drifter climatology including data from March 1995 to March 2009 (Lumpkin and Pazos, 2006) also confirms the good representation of the near-surface velocity from HYCOM data."

4) The manuscript states a lack of observational data, but several circulation studies have been done in this region that are not acknowledged in this manuscript. The manuscript would benefit from a more detailed description of how model output compare to both YoMaHa observations as well as previously published observations. Suggestions here: Zhankun Wang et al 2014 (Deep Sea Research) Seasonal and annual variability of vertically migrating scattering layers in the northern Arabian Sea. Zhankun Wang et al 2013 (Deep Sea Research) High salinity events in the northern Arabian Sea and Sea of Oman. Sarah Stryker Vitale et al 2017 (Dynamics of Atmospheres and Oceans) Circulation analysis in the northwest Indian Ocean using ARGO floats and surface drifter observations, and SODA reanalysis output

Thanks for pointing out these publications. The analysis of near-surface circulation in the northwestern Indian Ocean based on drifter data by Vitale et al. (2017) is used now for the validation of the HYCOM velocity data. "Additionally, an analysis of seasonal surface velocities in the AS (Vitale et al., 2017; their Figs. 2a, 3a), which is based on a drifter climatology including data from March 1995 to March 2009 (Lumpkin and Pazos, 2006) also confirms the good representation of the near-surface velocity from HYCOM data." Zhankun Wang et al., 2013 and 2014 are now acknowledged in the introduction and the discussion. "In the northwestern part of the basin where the Gulf of Oman merges with the AS, PGW about constantly runs out throughout the year (Fig. 10c, d). Observations confirm the small seasonal variations of the PGW outflow (Johns et al., 2003), which can be influenced by cyclones (Wang et al., 2013)." "The seasonal changes significantly influence biogeochemical cycles, biological activity and ecosystem response (Hood et al., 2009; Resplandy et al., 2012; Brewin et al., 2012; Wang et al., 2014)." "Lagrangian particles cross regions with high primary production during the long-distance advection while looping around the northern part of the basin. A study of seasonal vertically migrating scattering layers reveals a rapid increase of biomass in the northern Arabian Sea in the layer between 250 and 450 m depth during the period of June to November (Wang et al., 2014)."

5) Page 9 Line 17- state approximate depths of isopycnals (here and elsewhere) in the text so the reader does not have to refer to the figure throughout

As the depth of the isopycnal surfaces are variable, we decided to refer to the density in the text, because this is the more precise information. To make it easier for the reader to follow, we now added the depth of the isopycnal surface at a few passages in the text: We added the approximate depth in the manuscript where you suggested it: "Lagrangian trajectories were calculated on basis of the daily HYCOM reanalysis velocities on the isopycnal surface of 27 kg/m3 lying in the depth range of 450 to 500 m." We also added the depth to the caption of Fig.2 for better reading: "Mean seasonal velocity for the Arabian Sea based on HYCOM data on the isopycnal surface of $\sigma$=27

kg/m3 in the depth range of 450 to 500 m for..." Further we changed Fig. 3 so that the density layers are directly in the figure. We hope that makes it easier for the reader to follow.

6) The last sentence of the conclusion is not clearly written.

Thank you for that comment, we changed the last paragraph to: "The seasonal variability of advective pathways into the ASOMZ agrees well with the weak seasonal oxygen cycle and shows clear differences between the eastern and western basin. Still the oxygen content of advected water masses is strongly influenced by the strength and seasonality of biogeochemical processes in the AS. Nonetheless, we conclude that the advection of water mass plays a crucial role for the eastward shift of the ASOMZ and might also be responsible for the maintenance of low oxygen in the ASOMZ throughout the year. However, we cannot state whether physical or biogeochemical processes play the dominating role for the seasonal variability of the ASOMZ based on this method."

Technical Comments 7) Page 4 line 12 "it's" should be "its" Thank you for noticing that. We changed that.

8) Page 8 Line 25- Not clear to me what "lack of 3 month" means- typo?

We apologise the misunderstanding. Lack of 3 month was meant to describe a temporal offset in time. We hope it is better to understand with the new sentence: "Seasonal differences in particle movement around the release locations can be predicted by starting the calculations with a temporal offset of 3 month within the same year (January, April, July and October)."

9) Page 14 Line 27- "weak" misspelled Thanks, we changed that.

Please also note the supplement to this comment:
https://www.ocean-sci-discuss.net/os-2020-9/os-2020-9-AC2-supplement.pdf

**Fig. 1.** Figure 3: Annual mean of dissolved oxygen concentration along 62°E (left), 66.5° E (right) and 19° N (middle) from the WOA 13 climatology (see Figure 1). Advective pathways from Lagrangian particles a

[Figure]

**Fig. 2.** Figure 6: Exemplary advective Lagrangian pathways connecting the Persian Gulf (blue) and the Red Sea (red) with the (a) western release (WR) and the (b) eastern release (ER) locations marked in black.

[Figure]

**Fig. 3.** Figure 11: Schematic of the main advective pathways of the water masses for the a) winter, b) summer, c) spring inter- and d) autumn inter-monsoon seasons. See also Figure 1. The dark green dots indic

---

## Author Response (AR2)

**Suggestions for revision by Referee #3 and point-by-point response**

In this manuscript, the seasonal variability of the oxygen minimum zone in the Arabian Sea was studied based on particle
trajectory experiments. The results of the particle pathways and transit time analyses provide some convincing evidence of
the basin scale circulation at intermediate depth and its connection to the seasonal dynamics of the OMZ. Overall, this study
is interesting and may have some relevance to the mechanism of OMZ in other upwelling regions. However, I believe this
paper require a proper revision before it can be considered for publication. First, the title of the paper is misleading. The
linkage of the circulation should be related to a specific aspect of the OMZ.

*You are right. We specified the title to prevent misunderstandings:*
*Seasonal variability of the Arabian Sea intermediate circulation and its impact on seasonal changes of the upper Oxygen*
*Minimum Zone*

Second, the validation of the HYCOM circulation at the isopycnal surface need be include. Although a validation of the
surface current has been given in the MS, it does not prove an accuracy of the subsurface flow. Actually, it may be obtained
simply based on the drifts of Argo in the interior of the ocean.

*Due to the lack of observations at intermediate depths a validation of the circulation at the isopycnal surface is difficult.*
*Argo floats provide circulation data in 1000m depth. However, comparing different previous studies on velocities in 1000m*
*depth from Argo floats show high discrepancies that make it hard to validate the model in those depth (compare e.g.*
*YoMaHa'07 and Vitale et al., 2017). Those discrepancies arise from the sparse data coverage (101 floats between 2002 –*
*2009; Vitale et al., 2017) compared to the surface, where all studies are in agreement. Still, the main features are covered*
*by the model as seen in the new figure S2 in the supplement and mentioned in the manuscript:*
*"For a brief validation of the HYCOM velocity data in a depth of 1000 m (Fig. S2) we compared the HYCOM climatology*
*with the circulation described in Vitale et al. (2017). Their study is similarly based on Argo data but contrary to the*
*YoMaHa'07 climatology they only provide velocity data when a minimum of three velocity vectors are present and do*
*provide a variance ellipse at each point. The north–eastward flow along the coast of Somalia and Oman is also noticeable in*
*1000 m during the summer monsoon (Fig. S2b). Further offshore the deep reaching gyre circulation of the Great Whirl can*
*be identified. North of the Equator a uniform eastward flow is observed. This flow is connected to the EACC that separates*
*from the coast while crossing the equator (Vitale et al., 2017). During the winter monsoon, the flow at about 5°N changes*
*direction into a westward flow. This flow splits at the Somali coast into a strong southward and a weaker northward flow*
*(Fig. S2a). Also seen in Vitale et al. (2017) and the HYCOM velocity data is a westward flow in the Gulf of Aden during*
*wintertime. The northeastern AS has comparable weak velocities in 1000 m depth for both monsoon seasons."*

Finally, a schematic plot may be necessary to summarize the complex mechanism that controls the seasonal variability of the
OMZ following the long discussion.

*We made an update of Fig. 11 to summarize relevant processes that have an impact on the seasonal variability of the OMZ.*
*The schematic plot includes circulation pathways for every season and areas of strong primary production as well as*
*monsoon wind direction, upwelling areas and extension of the OMZ. We use the schematic plot throughout the discussion to*
*make it easier for the reader to follow.*

In addition, the English writing of the MS need be improved a lot. I found some sentences are very hard to understand and others contain obvious errors in grammar or typos. These issues appear throughout this MS.

*We had the whole MS proofread for typos and grammar errors by and hope that the English writing improved and is better to understand now.*

*We had the whole MS proofread for typos and grammar errors by and hope that the English writing improved and is better to understand now.*

[revised manuscript text omitted]